# PRDM9 activity depends on HELLS and promotes local 5-hydroxymethylcytosine enrichment

Yukiko Imai[1†‡], Mathilde Biot[1†], Julie AJ Clément[1], Mariko Teragaki[1], Serge Urbach[2], Thomas Robert[1§], Frédéric Baudat[1], Corinne Grey[1]*, Bernard de Massy[1]*

[1]Institut de Génétique Humaine (IGH), Centre National de la Recherche Scientifique, Univ Montpellier, Montpellier, France; [2]Institut de Génomique Fonctionnelle, Université de Montpellier, CNRS, INSERM, Montpellier, France

*For correspondence:
corinne.grey@igh.cnrs.fr (CG);
bernard.de-massy@igh.cnrs.fr
(BM)

[†]These authors contributed equally to this work

Present address: [‡]Department of Gene Function and Phenomics, National Institute of Genetics, Mishima, Japan; [§]Centre de Biochimie Structurale (CBS), CNRS, INSERM, Univ Montpellier, Montpellier, France

**Abstract** Meiotic recombination starts with the formation of DNA double-strand breaks (DSBs) at specific genomic locations that correspond to PRDM9-binding sites. The molecular steps occurring from PRDM9 binding to DSB formation are unknown. Using proteomic approaches to find PRDM9 partners, we identified HELLS, a member of the SNF2-like family of chromatin remodelers. Upon functional analyses during mouse male meiosis, we demonstrated that HELLS is required for PRDM9 binding and DSB activity at PRDM9 sites. However, HELLS is not required for DSB activity at PRDM9-independent sites. HELLS is also essential for 5-hydroxymethylcytosine (5hmC) enrichment at PRDM9 sites. Analyses of 5hmC in mice deficient for SPO11, which catalyzes DSB formation, and in PRDM9 methyltransferase deficient mice reveal that 5hmC is triggered at DSB-prone sites upon PRDM9 binding and histone modification, but independent of DSB activity. These findings highlight the complex regulation of the chromatin and epigenetic environments at PRDM9-specified hotspots.

## Introduction

In sexual reproduction, genetic information from both parental genomes is reassorted through chromosome segregation during meiosis. Additional genetic diversity is generated by recombination events between parental homologous chromosomes (homologs) that take place during the first meiotic prophase. Meiotic recombination leads to reciprocal (crossover) and non-reciprocal (gene conversion) transfer of genetic information. Crossovers establish physical links between homologs that are maintained until the end of prophase of the first meiotic division and are essential for the proper segregation of homologs at the first meiotic division. Gene conversion without crossovers promotes interactions between homologs, thus mechanistically contributing to the proper execution of the crossover pathway. Gene conversion also leads locally to the replacement of small regions (typically, from a few to a few hundred base pairs) from one parental genome to the other (*Chen et al., 2007*). Therefore, meiotic homologous recombination enhances genetic diversity (*Coop and Przeworski, 2007*) and is essential for fertility (*Hunter, 2015*). Homologous recombination events are generated by the programmed induction of DNA double-strand breaks (DSBs) followed by their repair, as a crossover or not, with a chromatid from the homolog (*Baudat and de Massy, 2007*). Meiotic DSBs are tightly controlled in time, space, and frequency to drive the homologous recombination pathway while keeping genome integrity (*de Massy, 2013*; *Keeney et al., 2014*; *Sasaki et al., 2010*).

Remarkably, two distinct pathways control DSB localization (*Lichten and de Massy, 2011*). In several species, including plants and fungi, DSBs occur at promoter regions or regions of accessible chromatin without detectable sequence specificity. This pattern of DSB localization has been

thoroughly analyzed in *Saccharomyces cerevisiae* (*Pan et al., 2011*). The DSB machinery, which involves several proteins including Spo11 that carries the catalytic activity for DNA cleavage (*de Massy, 2013*), is somehow directed to such accessible chromatin sites. Several factors that act locally, such as chromatin structure, but also higher order chromosome organization features are important for DSB formation at these accessible regions (*Lam and Keeney, 2015*). In contrast, in humans and mice, and possibly in some non-mammalian species (*Baker et al., 2017*), DSBs occur at sites bound by PRDM9 and not at promoters (*Pratto et al., 2014*; *Smagulova et al., 2011*). The *Prdm9* gene, which is present in a wide range of metazoans, is expressed specifically in meiocytes, at the stage of meiotic DSB formation. It encodes a protein that has a sequence-specific DNA- binding domain with multiple potential targets in the genome. PRDM9 also has a PR-SET domain with methyltransferase activity and promotes the tri-methylation of lysine 4 (H3K4me3) and of lysine 36 (H3K36me3) of histone H3 on nucleosomes adjacent to PRDM9-binding sites (*Grey et al., 2018*). This methyltransferase activity is essential for DSB formation at PRDM9 sites (*Diagouraga et al., 2018*). In mice lacking PRDM9, DSBs form at promoters and enhancers (*Brick et al., 2012*; *Mihola et al., 2019*).

The various steps that take place from PRDM9 binding to DSB formation are still poorly understood. Specifically, it is not known how the DSB machinery is recruited or activated, and how the different molecular steps proceed in a local chromatin environment that is a priori not specified before PRDM9 binding. Moreover, upon binding, PRDM9 promotes chromatin modifications with the deposition of H3K4me3 and H3K36me3 on adjacent nucleosomes (*Baker et al., 2014*; *Grey et al., 2017*; *Powers et al., 2016*). One or both of these modifications are predicted to be required for DSB activity, because PRDM9 methyltransferase is essential for DSB activity at PRDM9 sites (*Diagouraga et al., 2018*), unless another substrate of PRDM9 methyltransferase is involved. In addition, these histone modifications may play a role in DSB repair. Indeed, ZCWPW1, a protein reader of H3K4me3 and H3K36me3, is required for efficient DSB repair (*Huang et al., 2020*; *Li et al., 2019*; *Mahgoub et al., 2020*; *Wells et al., 2019*). Whatever their exact role, the presence of both histone modifications leads to a unique chromatin landscape at PRDM9 sites that is not present at promoters, where H3K36me3 is depleted (*Grey et al., 2017*; *Lam et al., 2019*; *Powers et al., 2016*). Therefore, the specific chromatin environment at PRDM9 sites may be important for DSB activity and repair. In mice lacking PRDM9, where DSBs form mainly at promoters and enhancers, an inefficient DSB repair is observed (*Brick et al., 2012*; *Hayashi and Matsui, 2006*), which could be due to the chromatin environment at those sites, and/or alternatively to improper regulation of DSB formation.

Other histone modifications have been identified at PRDM9 sites (*Buard et al., 2009*; *Davies et al., 2016*; *Lam et al., 2019*). It has been proposed that epigenetic features at DSB hotspots are also responsible for some of the observed DSB activity differences in male and female mouse meiosis (*Brick et al., 2018*). Indeed, besides chromatin, the global cytosine methylation level is different between sexes: low in prophase oocytes (*Kagiwada et al., 2013*; *Seisenberger et al., 2012*) and high in spermatocytes (*Gaysinskaya et al., 2018*). Cytosine methylation is a dynamic epigenetic modification that can be removed by the actions of ten-eleven-translocation (TET) enzymes, the first product of which is 5-hydroxymethylcytosine (5hmC) (*Tahiliani et al., 2009*). Interestingly, a local increase in 5hmC has been detected at PRDM9 sites in mouse spermatocytes (*Brick et al., 2018*), indicative of another potential layer of modification at DSB sites.

To gain insights into these molecular steps and to identify proteins involved at PRDM9-binding sites, we developed a proteomic approach to identify PRDM9 partners. We found that HELLS, a member of the SNF2-like family of chromatin remodelers, interacts with PRDM9. This interaction has also been recently shown by Spruce and colleagues (*Spruce et al., 2020*). Interestingly, HELLS was previously implicated in the regulation of DNA methylation, transposable element expression, heterochromatin dynamics, and DSB repair in somatic cells (*Burrage et al., 2012*; *Dennis et al., 2001*; *Kollárovič et al., 2020*; *Lungu et al., 2015*; *Yu et al., 2014b*). HELLS is also required for female and male meiosis (*De La Fuente et al., 2006*; *Zeng et al., 2011*). A recent study showed that HELLS is needed for proper meiotic DSB localization and acts as a PRDM9-dependent chromatin remodeler of meiotic hotspots (*Spruce et al., 2020*). Here, we found that in mouse male meiosis, HELLS is directly involved in the control of DSB activity by ensuring PRDM9 binding and thus DSB formation at PRDM9-dependent sites, consistent with the results obtained by *Spruce et al., 2020*. This HELLS activity appears to solve the challenge of chromatin accessibility for PRDM9 binding. We also show

that HELLS-dependent PRDM9 binding and PRDM9 methyltransferase activity are required for efficient 5hmC enrichment at PRDM9-binding sites. This epigenetic modification is a feature of PRDM9-specified hotspots that is not dependent on DSB formation.

## Results

### HELLS interacts with PRDM9

To identify proteins that interact with PRDM9 we first chose to express a tagged version of human PRDM9 in HeLa S3 cells, which do not express PRDM9 (*Morin et al., 2008*), and performed immunoprecipitation (IP) followed by mass spectrometry analysis allowing us to identify candidates that are not germline-specific. We generated two stable cell lines that express the human *PRDM9*[A] allele (*Baudat et al., 2010*) with an epitope tag (FLAG-HA) inserted at the amino- (Nt) or carboxy-terminal (Ct) end (see Materials and methods). Unmodified HeLa S3 cells were used as negative control. We purified tagged PRDM9[A]-containing complexes from HeLa S3 cell nuclear extracts by FLAG affinity, followed by HA affinity purification (*Table 1—source data 1*).

First, we carried out pilot experiments by selecting proteins with a size that ranged between 70 and 80 kD and between 95 and 120 kD after silver staining to potentially identify PRDM9 and other proteins, respectively. PRDM9 peptides were the first and the third most abundant peptides for Nt- and Ct-tagged PRDM9, respectively, only in the 70–80 kD size range (*Table 1*). Although the predicted molecular weight (MW) of tagged PRDM9 is 103 kD, its detection in the 70–80 kD size range is compatible with its faster than predicted migration during denaturing gel electrophoresis (see western blots in *Table 1—source data 1*). HELLS peptides were the first and the second most abundant peptides for Nt- and Ct-tagged PRDM9, respectively, only in the 95–120 kD size range, in agreement with HELLS predicted MW (97 kD) (*Jarvis et al., 1996*). Then, we repeated the experiments, but without size selection and by analyzing the full protein content after affinity purification. This analysis confirmed the pilot experiment findings, and highlighted HELLS as a major PRDM9 partner. In this condition, with both Nt- and Ct-tagged PRDM9, HELLS was the first in the list of proteins identified by mass spectrometry and ranked by peptide abundance. We did not detect HELLS peptides in IP experiments from HeLa S3 cells without the PRDM9-expressing vector. The PRDM9 and HELLS peptide counts, and protein coverages are shown in *Table 1* (see *Supplementary file 1* for the full list of proteins). We did not perform any other analysis or quantification of the proteome present in these samples for this study.

As HELLS is expressed in gonads and is essential for gametogenesis (*De La Fuente et al., 2006*; *Zeng et al., 2011*), we then tried but failed to detect any interaction between HELLS and PRDM9 by western blotting after IP of mouse testis protein extracts. This could be due to technical problems linked to the used antibodies since this interaction was recently detected in mouse testis extracts by *Spruce et al., 2020*. Therefore, we used mass spectrometry after IP with a polyclonal antibody against PRDM9 or normal rabbit serum (mock) (*Table 1*, *Table 1—source data 1*). The relative abundance of HELLS peptides was lower in the assays with mouse testis extracts compared with HeLa S3 cell extracts, partly due to higher noise. Nevertheless, HELLS peptides were enriched in extracts purified with the anti-PRDM9 antibody, compared with mock control. In three independent experiments, 14, 6, and 7 HELLS peptides were obtained after IP with the anti-PRDM9 antibody, and 6, 5, and one in the mock controls (*Table 1*). HELLS enrichment in IP experiments with the anti-PRDM9 antibody was also revealed by quantification based on the Label-Free-Quantification ranks (*Table 1* and *Supplementary file 1*).

As a complementary approach to proteomics, we searched for PRDM9 partners by yeast two-hybrid screening. Using mouse PRDM9 without zinc fingers as bait and a mouse juvenile testis cDNA bank, we identified six clones that corresponded to HELLS, indicating a direct interaction between PRDM9 and HELLS. All six clones shared a domain that included residues 30 to 448 of HELLS (*Figure 1*). To better map the HELLS region involved in the interaction with PRDM9, we generated different HELLS constructs and found that the C-terminal region (569-821) of HELLS was dispensable for this interaction (*Figure 1B*). We could not detect any interaction with PRDM9 upon deletions at the N-terminus or C-terminus of the 1–569 region, such as in the HELLS constructs 193–569 and 1–408, respectively. This suggests the potential involvement of the N-terminal (1-193) and of the

**Table 1.** HELLS is co-immunoprecipitated with PRDM9.

Two independent immunoprecipitation experiments were performed using HeLa cells and mouse testis extracts. In the first experiment, HeLa S3 cells that express N-terminally (Nter) or C-terminally (Cter) tagged human PRDM9 or without PRDM9 expression vector (no PRDM9) were used to identify proteins that interacts with PRDM9 after size selection (95–120 kD and 70–80 kD), and without size selection. Mouse testis extracts were prepared without (rep1) or after incubation with benzonase (rep2) (in duplicate). IP were performed with an anti-PRDM9 antibody or with normal rabbit serum (mock). For each protein (PRDM9 and HELLS), the total number of peptides, the protein rank in the whole set of proteins with at least one peptide, and ranked by number of peptides, and the sequence coverage are indicated; na: not applicable. For mouse testis extracts, the rank difference of the label free quantification intensity (LFQ) between IPs with anti-PRDM9 and mock are indicated. The full list of the identified peptides is in *Supplementary file 1*. Extracts analysis by electrophoresis are presented in *Table 1—source data 1*.

| IP | Total peptides PRDM9 | Rank PRDM9 | Sequence coverage PRDM9 (%) | Total peptides HELLS | Rank HELLS | Sequence coverage HELLS (%) |
|---|---|---|---|---|---|---|
| HeLa with size selection | | | | | | |
| HeLa PRDM9 Nter 95–120 KD 70–80 KD | 0 7 | na 1/21 | na 6.4 | 11 0 | 1/16 na | 11.9 na |
| HeLa PRDM9 Cter 95–120 KD 70–80 KD | 0 7 | na 3/34 | na 7.3 | 4 0 | 2/32 na | 4.4 na |
| HeLa no PRDM9 95–120 KD 70–80 KD | 0 0 | na | na | 0 0 | na | na |
| HeLa without size selection | | | | | | |
| HeLa PRDM9 Nter | 38 | 6/447 | 29.4 | 97 | 1/447 | 48.1 |
| HeLa PRDM9 Cter | 35 | 4/364 | 33.7 | 44 | 1/364 | 37.6 |
| HeLa no PRDM9 | 0 | na | na | 0 | na | na |
| Mouse testis rep1 | | | | | | |
| IP PRDM9 | 24 | 27/571 | 35.1 | 14 | 75/571 | 24.1 |
| mock | 1 | 538/571 | 1.2 | 6 | 211/571 | 9.3 |
| LFQ Rank difference | | 441 | | | 113 | |
| Mouse testis rep2 (+benzonase) | | | | | | |
| IP PRDM9 | 14 15 | 39/890 41/948 | 26.3 | 6 7 | 187/890 178/948 | 11 |
| mock | 1 0 | 782/890 na | 1.4 | 5 1 | 323/890 506/948 | 7.2 |
| LFQ Rank difference | | 870 688 | | | 122 468 | |

The online version of this article includes the following source data for Table 1:

Source data 1. Purification of protein complexes.(A) Western blot analysis after complex purification by Flag-HA of extracts from HeLa S3 cells. HeLa S3 cells without PRDM9 expression vector, or expressing human PRDM9 tagged with Flag-HA at the C-terminus (PRDM9-Ct) or N-terminus (PRDM9-Nt) were used. Protein fractions of the extracts before IP (S1: cytoplasmic fraction, S2: nuclear fraction as input for IPs, ppt: insoluble pellet) and after the affinity purification steps were analyzed by western blotting using an anti-PRDM9 antibody. (B) Analysis of affinity-purified proteins after silver staining (sample without size selection). Eluates 1, 2 and resin fractions obtained from affinity purification (HA) of extracts initially prepared from HeLa S3 cells without PRDM9 expression vector (M), or expressing human PRDM9 tagged with Flag-HA at the C-terminus (Ct) or at the N-terminus (Nt) were separated by electrophoresis and silver stained. Mixtures of Eluate 1 and 2 were used for mass spectrometry analysis. (C) Western blot analysis of complex purification using an anti-PRDM9 antibody and mouse testes extracts (Mouse testis rep1). Protein extracts obtained during the Dignam-based purification (S1: cytoplasmic fraction, S2: nuclear fraction, S3: DNase-treated, and ppt: pellet) were loaded. Input (S2), unbound (UB), and proteins immunoprecipitated (IP) with an anti-PRDM9 antibody or normal rabbit serum (mock) were analyzed by western blotting. Detection was with an anti-PRDM9 antibody. Loading: 1 and 10% of input and IP samples, respectively. (D) Analysis of affinity-purified proteins by silver staining (Mouse testis rep1). Input, and samples IP with an anti-PRDM9 antibody or with normal rabbit serum (mock) were loaded. Bovine serum albumin (BSA) was used as control. Proteins were separated by electrophoresis and silver stained. (E) Western blot analysis of complex purification using an anti-PRDM9 antibody in extracts from mouse testes incubated with benzonase (Mouse testis rep 2), in duplicate (a and b). Protein extracts obtained during the Dignam-based purification steps (S1: cytoplasmic fraction, S2: nuclear fraction, S3: DNase-treated, and ppt: pellet) were loaded. Input (S2) and proteins IP with an anti-PRDM9 antibody or rabbit serum (mock) were analyzed by western blotting. Detection was with an anti-PRDM9 antibody. Loading: 1% and 10% of input and IP samples. (F) Analysis of affinity purified proteins by silver staining (Mouse testis rep2). Input, and samples IP with an anti-PRDM9 antibody or with normal rabbit serum (mock) were loaded. BSA was used as control. Proteins were separated by electrophoresis and stained with silver.

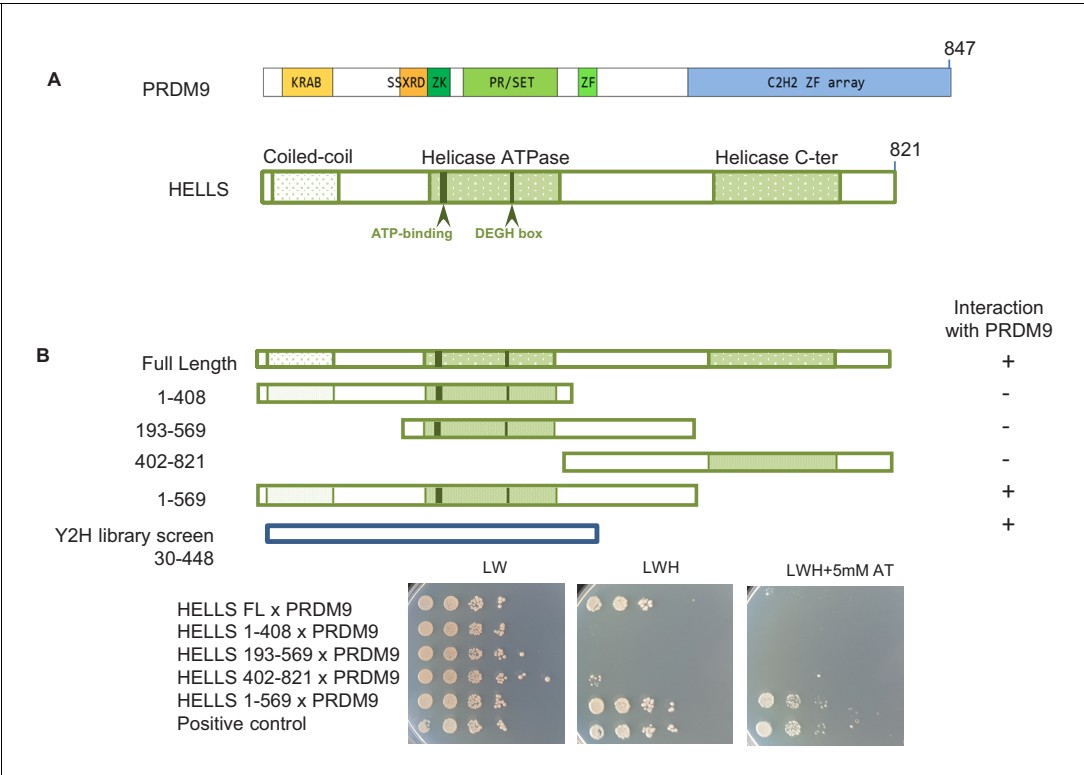

**Figure 1.** HELLS interacts with PRDM9. (**A**) Domains of PRDM9 and HELLS PRDM9 includes a Krüppel-associated box domain (KRAB), a synovial sarcoma-X break point-repression domain (SSXRD), a Su(var)3–9, Polycomb-group protein enhancer of zeste and trithorax-group protein TRX domain (PR/SET) that is preceded and followed by zinc-finger domains (ZK and ZF, respectively), and a C2H2-type zinc-finger array (C2H2 ZF array). HELLS contains a coiled-coil domain, a helicase ATPase domain, and a helicase C-terminal domain. (**B**) Interaction between PRDM9 and HELLS by yeast two-hybrid assays. Full-length and four fragments of mouse HELLS were used to test for interaction with mouse PRDM9 (full length). HELLS domains were fused to the Gal4 activation domain (GAD), and PRDM9 was fused to the Gal4 DNA-binding domain (GBD). A positive interaction was detected for full-length HELLS and fragment 1–569. Growth was tested on medium without leucine and tryptophan (LW), without leucine, tryptophan and histidine (LWH), and without leucine, tryptophan and histidine with 5 mM amino-triazole (LWH + 5 mM AT). A diploid strain that expresses pGAD-REC114 and pGBD-MEI4 (*Kumar et al., 2010*) was used as positive control. The HELLS region of the cDNAs isolated by yeast two-hybrid screening is shown. Controls are shown in *Figure 1—figure supplement 1*.

The online version of this article includes the following figure supplement(s) for figure 1:

**Figure supplement 1.** Controls for yeast two-hybrid assays.

central (408-569) regions of HELLS in the interaction with PRDM9 (*Figure 1B* and *Figure 1—figure supplement 1*).

## HELLS is required for PRDM9-dependent meiotic DSB formation

To evaluate the functional role of HELLS in meiotic recombination, we generated a mouse line in which *Hells* was conditionally ablated only during male meiosis by meiotic-specific expression of CRE under the control of the *Stra8* promoter on a transgene (*Sadate-Ngatchou et al., 2008*) (*Hells* cKO) (*Figure 2—figure supplement 1*), because HELLS is essential for mouse development (*Geiman et al., 2001*). A previous study where HELLS-deficient testes were analyzed by allografting embryonic tissues showed that HELLS is required for meiotic progression during spermatogenesis (*Zeng et al., 2011*). A potential interpretation for this meiotic defect was that alterations of DNA methylation in the absence of HELLS may affect, directly or indirectly, some properties of meiotic prophase and synapsis between homologous chromosomes (*Zeng et al., 2011*). This phenotype shares some similarity with the defects observed in *Hells* KO oocytes (defects in meiotic recombination and homologous synapsis, and changes in DNA methylation at repetitive DNA elements and pericentric heterochromatin) (*De La Fuente et al., 2006*).

Therefore, to test whether HELLS is directly implicated in meiotic recombination, we first precisely determined the meiotic stage(s) and steps that could be affected by HELLS depletion. In this study, we used various mouse strains with wild-type *Hells* alleles (*Hells*$^{fl/+}$, *Hells*$^{fl/+}$ *Stra8-Cre*$^{Tg}$, *Hells*$^{fl/-}$) that are all named *Hells* CTRL hereafter. Meiotic-specific *Hells* mutant mice were *Hellsfl*$^{fl/-}$*Stra8-Cre*$^{Tg}$ and are named *Hells* cKO hereafter.

Western blot analysis of wild-type mouse testis whole cell extracts showed that HELLS protein could be detected from 4 days post-partum (dpp) to 15 dpp and in adults. PRDM9 was detected from nine dpp when cells have entered meiosis, but not at 4 and 6 dpp before meiosis entry (*Figure 2—figure supplement 2*). In testis nuclear extracts from 22 dpp *Hells* cKO animals, HELLS level was greatly reduced (*Figure 2A*), but not the nuclear PRDM9 protein level. The residual HELLS protein expression in testis nuclear extracts from *Hells* cKO mice might be due to incomplete CRE-induced deletion of *Hells* in some spermatocytes, as suggested by the cytological analysis presented below. We analyzed HELLS staining by immunofluorescence on spread spermatocytes of adult *Hells* CTRL and *Hells* cKO mice (*Figure 2B*). In *Hells* CTRL nuclei, we could detect HELLS as punctuate staining that covered nuclear chromatin, with the highest intensity at leptotene and zygotene and absence of specific staining at later stages (*Figure 2—figure supplement 3*). In *Hells* cKO nuclei, we did not detect HELLS staining in 75% of leptotene and zygotene nuclei, but could observe a weak HELLS staining in about 25% of nuclei (not shown). Thus, in some *Hells* cKO spermatocytes, HELLS expression was not completely abolished, and the protein was still present in the nucleus. Histological analysis of *Hells* cKO mice revealed spermatogenesis defects with 89% of tubules without haploid cells (*Figure 2C,D*), suggesting an arrest of spermatocyte differentiation. The presence of 11% of tubules with some haploid cells might be explained by incomplete HELLS depletion in some spermatocytes. Moreover, the percentage of tubules with one or more TUNEL-positive cells was increased, indicative of apoptotic cells undergoing massive genomic DNA breakage (*Figure 2E* and *Figure 2—figure supplement 4*).

By immunostaining of spread spermatocytes, we showed that in *Hells* cKO mice, spermatocytes entered meiotic prophase and progressed until a pachytene-like stage with chromosomes only partially synapsed in most nuclei, whereas some nuclei had fully synapsed chromosomes (*Figure 2F*), consistent with previous observations on *Hells*-deficient spermatocytes (*Zeng et al., 2011*). We detected chromosome axes by the presence of the axial protein SYCP3, and synapses by the presence of the central element protein SYCP1. Ninety three percent of *Hells* CTRL nuclei that showed full-length axes were at the pachytene stage with 19 fully synapsed autosomes and a γH2AFX-positive chromatin domain containing the X and Y-chromosomes, called sex body. In contrast, only 13% of *Hells* cKO nuclei with full-length axes were similar to wild-type looking pachytene nuclei. This population of wild-type pachytene nuclei in *Hells* cKO mice could be due to incomplete depletion of HELLS in some spermatocytes, as discussed above.

Interestingly, in most *Hells* cKO spermatocytes, despite the normal level of nuclear PRDM9 detected by western blotting (*Figure 2A*), the PRDM9 signal detected by immunostaining was much reduced compared with wild-type (*Figure 2B*, *Figure 2—figure supplement 5*). As in nuclear spreads proteins that are not tightly bound to chromatin can be partially lost, this low PRDM9 signal in *Hells* cKO samples could indicate that in the absence of HELLS, PRDM9 localizes in the nucleus, but does not bind to chromatin efficiently. Overall, the DSB activity did not seem to be affected because we detected a large number of DMC1 foci. Conversely, DSB repair appeared to be defective, as indicated by the persistence of DMC1 and γH2AFX foci, and the absence of a normal XY sex body at the pachytene-like stage (*Figure 2G*).

To directly test DSB activity and localization in the absence of HELLS, we performed chromatin IP with an anti-DMC1 antibody followed by single-strand DNA sequencing (DMC1 ChIP-SSDS). DMC1 ChIP-SSDS allows recovering single-strand DNA bound by the strand exchange protein DMC1 (*Khil et al., 2012*). We performed these experiments in two wild-type (*Hells* CTRL) and two *Hells* cKO mice. Both wild-type and mutant mice express the PRDM9$^{Dom2}$ variant that binds to a specific set of genomic sites and promotes DSB formation at those sites (*Brick et al., 2012*; *Grey et al., 2017*). We detected 11133 and 17117 peaks of DMC1 enrichment in the *Hells* CTRL and *Hells* cKO samples, respectively. This indicated the presence of DSB activity in both genetic contexts, as observed by immunofluorescence. However, only 1129 peaks were common, representing 10% of *Hells* CTRL peaks, and 6.6% of *Hells* cKO peaks (*Figure 3A*, *Figure 3—figure supplement 1*). Analysis of the signal intensity in the genotype-specific peaks (*Hells* CTRL -specific and *Hells* cKO-specific)

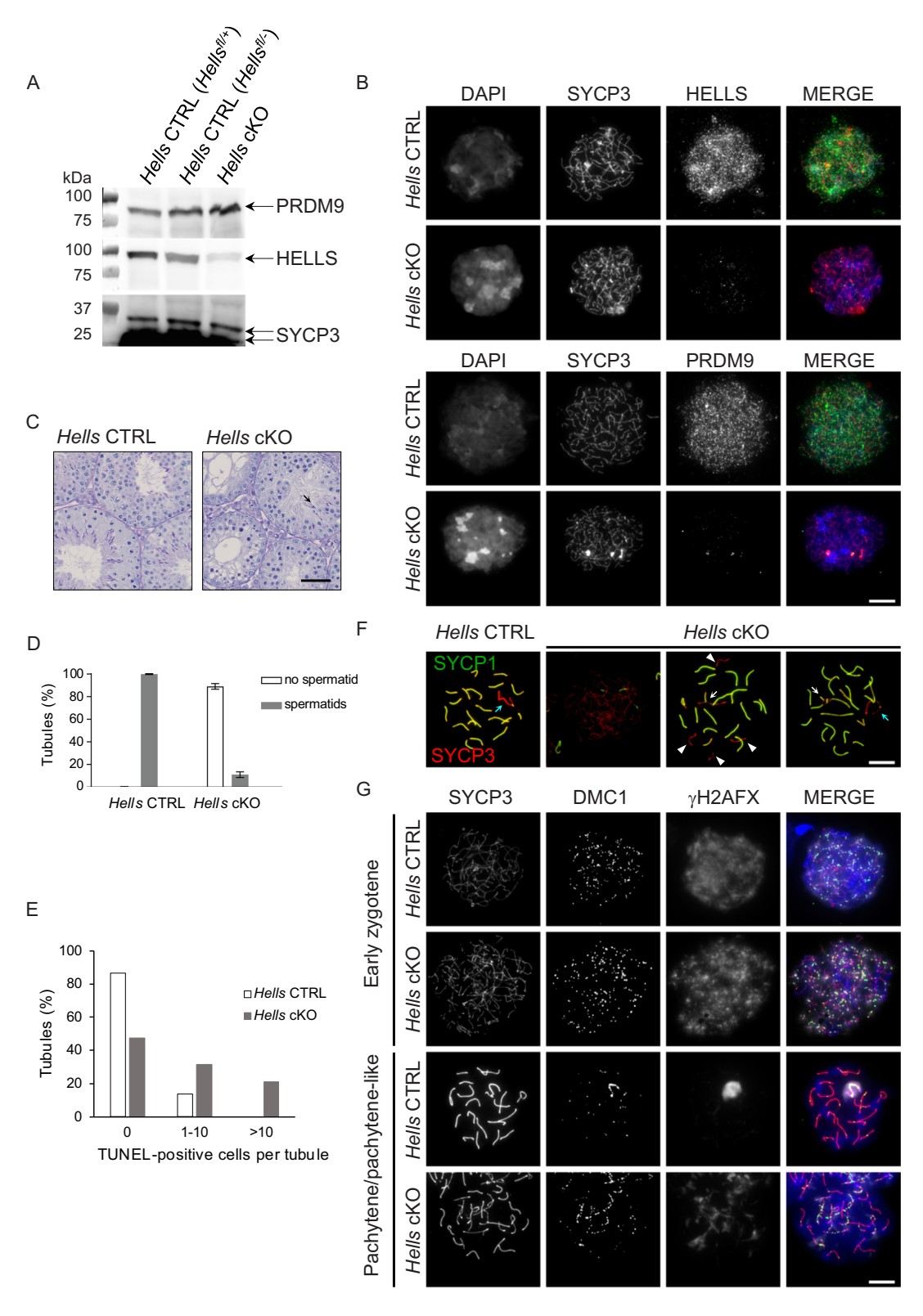

**Figure 2.** Meiotic prophase is defective in *Hells* cKO spermatocytes. (**A**) Detection of PRDM9, HELLS and SYCP3 expression in nuclear fractions of testes from *Hells* CTRL (*Hells*^fl/+ and *Hells*^fl/-) and *Hells* cKO mice at 22 dpp. *Hells* alleles are presented in *Figure 2—figure supplement 1*. HELLS and PRDM9 expression are presented in *Figure 2—figure supplement 2*. (**B**) Representative spreads of early zygotene spermatocyte nuclei from synchronized testes from *Hells* CTRL and *Hells* cKO mice after staining for DNA (DAPI, white or blue), SYCP3 (white or red) and HELLS (white or green)

*Figure 2 continued on next page*

*Figure 2 continued*

(top panels) or PRDM9 (white or green) (bottom panels). Anti-HELLS antibody from rabbit was used for these staining. Scale bar, 10 µm. HELLS and PRDM9 detection kinetics are presented in *Figure 2—figure supplements 3* and *5*. (**C**) Periodic acid-Schiff (PAS) staining of testis sections from 40 dpp *Hells* CTRL and *Hells* cKO mice. To visualize the glycoproteins/acrosomes (violet) and nuclei (blue), testis sections were stained with PAS and counterstained with hematoxylin. *Hells* CTRL testis sections (left panel) show normal spermatogenesis with well-organized stages of germ cell development, round spermatids with PAS-positive normal acrosomal caps, elongating and elongated spermatids. *Hells* cKO testis sections (right panel) show defective spermatogenesis with only few elongated spermatids (black arrow). Scale bar, 50 µm. (**D**) Proportions of seminiferous tubules without and with spermatids (mean ± SD) in testis sections from *Hells* CTRL and *Hells* cKO mice at 40 dpp. n = 4 testis sections from two mice. Data are available in *Figure 2—source data 1*. (**E**) Apoptosis detected by TUNEL assay in *Hells* CTRL and *Hells* cKO testes at 40 dpp. n = 2 testis sections from one mouse. TUNEL-positive cells are shown in *Figure 2—figure supplement 4*. Data are available in *Figure 2—source data 1*. (**F**) SYCP3 (red) and SYCP1 (green) staining of pachytene (*Hells* CTRL) and pachytene-like (*Hells* cKO) spermatocyte nuclei from 40 dpp mice. Arrowheads, unsynapsed chromosomes. White arrow, non-homologous synapsis. Blue arrows, sex chromosomes. Scale bar, 10 µm. (**G**) Representative spreads of early zygotene and pachytene or pachytene-like spermatocyte nuclei from 40 dpp *Hells* CTRL and *Hells* cKO mice, respectively, after staining for SYCP3 (white or red), DMC1 (white or green) and γH2AFX (white or blue). Scale bar, 10 µm.

The online version of this article includes the following source data and figure supplement(s) for figure 2:

**Source data 1.** Quantification of spermatid and TUNEL-positive sections.
**Figure supplement 1.** The *Hells* cKO allele.
**Figure supplement 2.** PRDM9 and HELLS protein levels during the first wave of spermatogenesis in wild-type mice.
**Figure supplement 3.** HELLS detection in *Hells* CTRL and *Hells* cKO spermatocytes Representative spreads of spermatocyte nuclei from *Hells* CTRL.
**Figure supplement 4.** TUNEL-positive cells detected in testis sections of control and *Hells* cKO mice.
**Figure supplement 5.** HELLS and PRDM9 detection in *Hells* CTRL and *Hells* cKO spermatocytes.

showed the absence of detectable signal in one genotype within peaks specific to the other genotype (*Figure 3B*). In the 1129 common peaks, the average DMC1 enrichment intensity was higher in *Hells* CTRL than in *Hells* cKO samples (*Figure 3B*). Among these common peaks, analysis of individual peak intensities revealed three subgroups, one subgroup with stronger intensity in *Hells* CTRL (n = 898 peaks), one subgroup with stronger intensity in *Hells* cKO (n = 154 peaks), and a smaller subgroup (n = 77 peaks) where the peak intensity was similar in both genotypes (*Figure 3—figure supplement 2A*). The group of 898 peaks with stronger intensity in *Hells* CTRL corresponded to DSB sites specified by PRDM9$^{Dom2}$. Indeed, an enrichment for H3K4me3 at these sites was observed specifically in the B6 strain that expresses PRDM9$^{Dom2}$, but not in the congenic RJ2 strain that expresses PRDM9$^{Cst}$, which binds to distinct genomic sites (*Figure 3—figure supplement 2B*). This suggests that these 898 peaks with stronger DMC1 intensity (in the cell population) in *Hells* CTRL may have a lower DSB level in *Hells* cKO, or may be active only in a smaller cell fraction in *Hells* cKO mice. We favor the second hypothesis, because our cytological analyses showed that HELLS is still detected in a small fraction of *Hells* cKO spermatocytes. The group of 154 peaks with higher DMC1 enrichment in *Hells* cKO were in regions with PRDM9-independent H3K4me3 enrichment (*Figure 3—figure supplement 2B*), suggesting a specific induction of DSB activity at these sites in the absence of HELLS. The group of 77 peaks with similar DMC1 intensity in *Hells* CTRL and *Hells* cKO showed a weak PRDM9-independent H3K4me3 enrichment. A low level of DSB activity at PRDM9-independent sites has been detected in mice that express PRDM9, and could account for these peaks (*Smagulova et al., 2016*).

To better understand the low overlap of DMC1 peaks in *Hells* CTRL and *Hells* cKO mice, we compared the *Hells* cKO peaks with those mapped in *Prdm9* KO mice. When PRDM9 is defective (such as in *Prdm9* KO mice) DSBs are formed at a different set of genomic sites, also called default sites. These sites overlap mainly with promoters and enhancers and are enriched in H3K4me3 (*Brick et al., 2012*). Remarkably, 85% of *Hells* cKO peaks overlapped with peaks detected in *Prdm9* KO mice (*Figure 3C and E*). This demonstrated that in the absence of HELLS, DSBs are no longer formed at PRDM9 sites, but are induced at default sites, similarly to what observed in *Prdm9* KO mice (*Figure 3E*). The lower number of peaks detected in *Hells* cKO samples (17117) compared with *Prdm9* KO mice (27732) could be due to a lower signal in *Hells* cKO samples. We hypothesized that mainly low intensity peaks in *Prdm9* KO mice should be undetectable in *Hells* cKO mice, and mainly high intensity peaks in *Prdm9* KO should be detected in *Hells* cKO mice, thus contributing to the population of the 14543 overlapping peaks. Indeed, among the peaks mapped in *Prdm9* KO samples, the peaks that were identified as overlapping with *Hells* cKO peaks were biased toward higher intensity compared with non-overlapping peaks (*Figure 3D*). The lower signal detected in *Hells* cKO

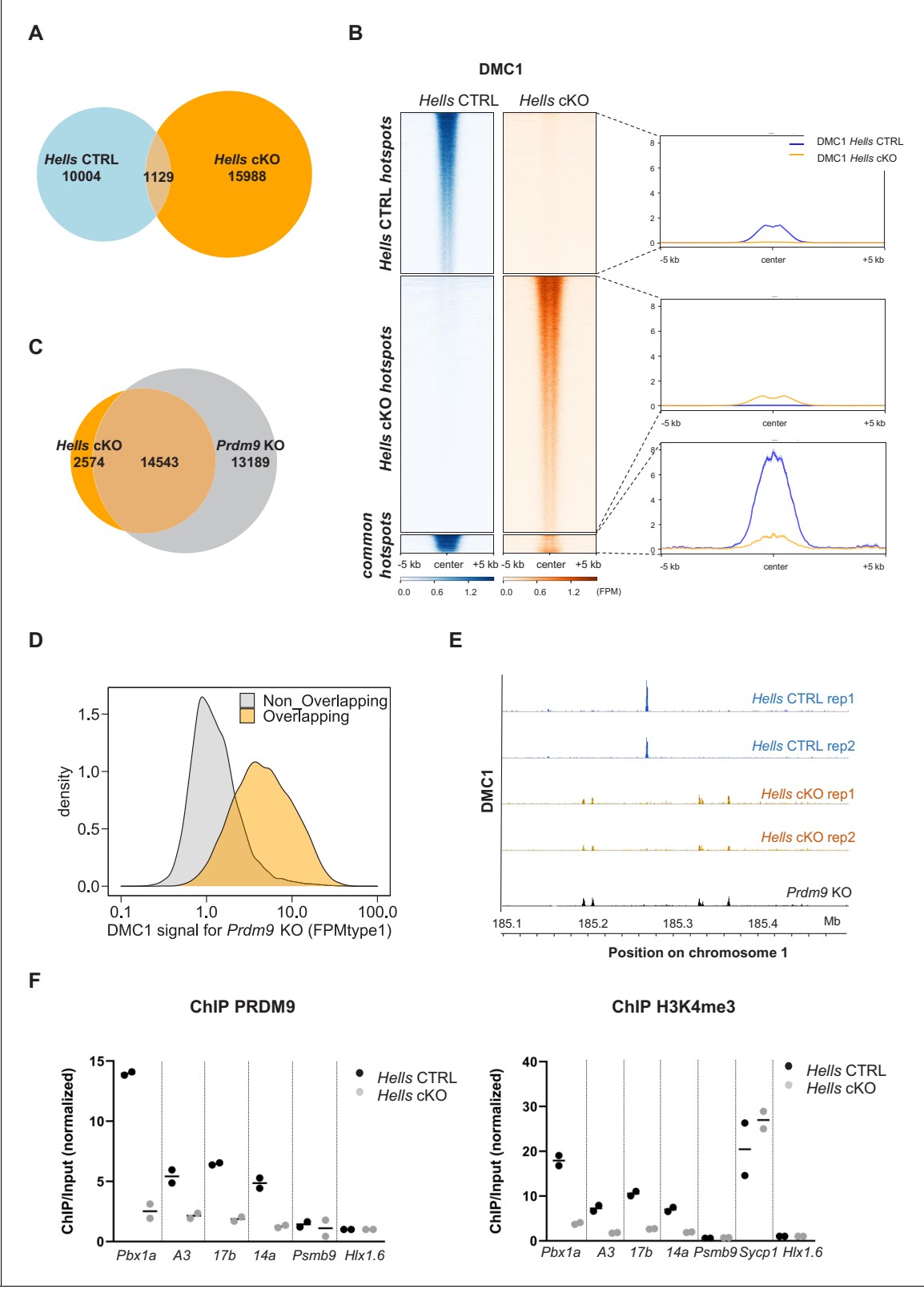

**Figure 3.** HELLS is required for the formation of meiotic DSBs at sites of PRDM9-dependent DSB formation; DSBs are directed at default sites in the absence of HELLS. (A) Limited overlapping between DSB hotspots from *Hells* CTRL and *Hells* cKO testis samples. Only hotspot centers (DMC1-SSDS peaks) that overlapped within a 400bp-window were considered as common. The others were considered as *Hells* CTRL- or *Hells* cKO-specific hotspots. Controls are shown in *Figure 3—figure supplement 1*. (B) Distribution of the DMC1-SSDS signal from *Hells* CTRL and *Hells* cKO testis samples around

*Figure 3 continued*

*Hells* CTRL, *Hells* cKO and common hotspots (as defined in (**A**)). The heatmaps show the DMC1-SSDS normalized fragments per million, calculated in a −5 kb to +5 kb window around hotspot centers and averaged within 10bp-bins. For the *Hells* CTRL - or *Hells* cKO-specific hotspots, the sites on the heatmaps were ranked by decreasing DMC1 intensity (from top to bottom) in the genotype where the peaks were detected. For the common hotspots, the sites were ranked by decreasing DMC1 intensity (from top to bottom) in *Hells* CTRL mice. The averaged profiles represent the mean DMC1-SSDS signal for each group. The analysis of common hotspots is shown in *Figure 3—figure supplement 2*. (**C**) Wide overlapping of DSB hotspots from *Hells* cKO and *Prdm9* KO testis samples. Hotspot (DMC1-SSDS peaks) centers that overlapped within a 400bp-window were considered as common. The others were considered to be *Hells* cKO- or *Prdm9* KO-specific hotspots. *Prdm9* KO data were from GSE99921 (*Brick et al., 2012*). (**D**) The DMC1-SSDS signal in *Prdm9* KO testis samples is either *Prdm9* KO-specific (i.e. not overlapping) or overlapping with *Hells* cKO-specific hotspots (as defined in (**C**)). Density of hotspot number is plotted as a function of the DMC1 signal in *Prdm9* KO mice, expressed as FPMtype1 (type1-single-strand DNA fragments Per Million of mapped reads, see Materials and methods and *Khil et al., 2012* for details). (**E**) DSB maps for *Hells* CTRL (blue) and *Hells* cKO (orange) testis samples (this study, two replicates for each genotype) and *Prdm9* KO testis samples (black, GSE99921; *Brick et al., 2012*) at a representative region of chromosome 1 (185.1Mb-185.5Mb). (**F**) Enrichment of PRDM9 and H3K4me3 is reduced at hotspots in *Hells* cKO compared with *Hells* CTRL samples. PRDM9 and H3K4me3 ChIP/Input ratios were calculated at several B6 (PRDM9$^{Dom2}$)-specific hotspots (*Pbx1a*, *14a*, *A3*, *17b*), at the *Sycp1* promoter (only for H3K4me3), and at two control regions that contain PRDM9$^{Cst}$-specific hotspots (*Psmb9.8* and *Hlx1.6*). All ratios were normalized to the ratios at *Hlx1.6*. At the four B6-specific hotspots, the difference between *Hells* cKO and *Hells* CTRL was statistically significant (two-tailed Mann-Whitney, p=0.0002). Data are available in *Figure 3—source data 1*.

The online version of this article includes the following source data and figure supplement(s) for figure 3:

**Source data 1.** PRDM9 and H3K4me3 ChIP-qPCR.
**Figure supplement 1.** DMC1 ChIP-SSDS reproducibility and controls.
**Figure supplement 2.** Common hotspots between *Hells* CTRL and *Hells* cKO testis samples.

could be explained by a difference in the sensitivity of the current ChIP-SSDS experiment and/or by a difference in DSB activity.

PRDM9-dependent DSB sites are characterized by several features that are implemented independently of DSB formation: PRDM9 binding, and enrichment for H3K4me3, H3K36me3 and H3K9ac on adjacent nucleosomes (*Buard et al., 2009*; *Davies et al., 2016*; *Grey et al., 2017*; *Lam et al., 2019*; *Powers et al., 2016*; *Spruce et al., 2020*). To identify the step of PRDM9-dependent DSB site designation affected by the absence of HELLS, we tested by ChIP-qPCR, PRDM9 binding and H3K4me3 levels at four representative PRDM9$^{Dom2}$ binding sites (*Pbx1a*, *A3*, *14a*, *17b*) that were used as reference in previous studies (*Billings et al., 2013*; *Diagouraga et al., 2018*). Enrichment for PRDM9 and for H3K4me3 were strongly reduced (at least by four-fold) at all four sites in *Hells* cKO spermatocytes compared with *Hells* CTRL cells (*Figure 3F*). This indicates that HELLS is required for efficient PRDM9 binding to its sites, consistent with the strong reduction in PRDM9 signal detected by immunofluorescence (*Figure 2B*). Moreover, this result provides a molecular interpretation for the absence of DSB activity at PRDM9-binding sites in *Hells* cKO spermatocytes.

## HELLS and PRDM9 are required for 5hmC enrichment at meiotic hotspots

Recently, it was shown that HELLS interacts with all three TET methylcytosine dioxygenases (*de Dieuleveult et al., 2020*; *Jia et al., 2017*). Upon oxidation, the activity of TET enzymes on methylated cytosines (5mC) leads to a first product, 5-hydroxymethylcytosine (5hmC). It has been proposed that 5mC conversion to 5hmC allows regulating 5mC levels for proper gene expression (reviewed in *Williams et al., 2012*). Unlike 5mC, 5hmC is globally associated with euchromatin and is depleted on heterochromatin in somatic cells (*Ficz et al., 2011*). Interestingly, in mouse male germ cells, 5hmC is enriched at some enhancers and promoters (*Gan et al., 2013*; *Hammoud et al., 2014*), and at meiotic DSB hotspots in pachytene spermatocytes (*Brick et al., 2018*).

To test whether 5hmC enrichment was correlated with the DNA-binding specificity of PRDM9, we took advantage of two congenic mouse strains (B6 and RJ2) that express PRDM9 variants with distinct DNA-binding specificities (PRDM9$^{Dom2}$ and PRDM9$^{Cst}$, respectively). In both strains, the sites of PRDM9 binding and activity have been mapped, and localize to distinct sets of genomic sites (*Grey et al., 2017*). As the mapping of PRDM9-dependent DSB hotspots can be done with different molecular approaches (ChIP with anti-PRDM9, -H3K4me3, or -DMC1 antibodies), we used the DMC1 ChIP-SSDS data that provide the optimal specificity and sensitivity, as reference for hotspots (*Grey et al., 2017*). We performed the 5hmC analysis using genomic DNA isolated from 95% pure

leptotene/zygotene cell populations (see Methods). In both B6 and RJ2 mouse strains, the 5hmC signal was correlated with DMC1 enrichment (*Figure 4A and B*), demonstrating that 5hmC enrichment depends on PRDM9 binding to its genomic targets.

The heatmaps of 5hmC enrichment at DMC1 sites revealed a correlation between the strength of the DMC1 hotspots and that of 5hmC (*Figure 4A*). We obtained similar results when the heatmaps were generated based on sites defined by PRDM9 ChIP (*Figure 4—figure supplement 1A*). We also noted that the mean 5hmC signal at hotspots was higher in RJ2 than in B6 samples (*Figure 4A*, *Figure 4—figure supplement 1A*). This correlated with the greater occupancy of the PRDM9$^{Cst}$ variant (expressed in RJ2 mice) compared with the PRDM9$^{Dom2}$ variant (expressed in B6 mice) (*Grey et al., 2017*). The 5hmC enrichment analysis and specifically the average enrichment plots showed a narrow distribution of the 5hmC enrichment that extended about +/- 250 bp from the peak center and overlapped closely with the enrichment profile of PRDM9 (*Figure 4C*). Peak centers were defined based on the DMC1 ChIP-SSDS signal and have been previously shown to overlap with PRDM9 DNA-binding motifs (*Smagulova et al., 2011*). However, 5hmC distribution was narrower than DMC1 distribution, which extends to the single-stranded DNA generated upon DSB end processing (*Figure 4—figure supplement 1B*). Moreover, the 5hmC maximum intensity was between the H3K4me3 peaks that delineate the positioned nucleosomes flanking the PRDM9 -binding sites (*Baker et al., 2014*; *Figure 4—figure supplement 1B*). Thus, 5hmC was predominantly taking place in the nucleosome-depleted region at and around PRDM9-binding sites.

Altogether, these findings suggest that 5hmC enrichment is functionally linked to PRDM9-binding activity. To directly test this hypothesis, we analyzed 5hmC in *Hells* cKO spermatocytes where PRDM9 binding to hotspots is defective (*Figures 2B* and *3F*). Strikingly, 5hmC enrichment at hotspots was lost in *Hells* cKO spermatocytes (*Figure 4D*). This suggests that 5hmC enrichment at meiotic hotspots is promoted by HELLS and/or PRDM9 binding, or by one of the subsequent steps depending on HELLS and PRDM9. Therefore, we tested whether PRDM9 methyltransferase activity was required, using a mouse strain (named B6-Tg(YF)) where two PRDM9 variants of distinct DNA-binding specificities are produced: the PRDM9$^{Dom2}$ variant with wild-type methyltransferase activity, and the PRDM9$^{Cst-YF}$ variant with defective methyltransferase activity due to a point mutation (Y357F) in the SET domain (*Diagouraga et al., 2018*; *Wu et al., 2013*). Our previous study established that the PRDM9$^{Cst-YF}$ variant binds to the binding sites of PRDM9$^{Cst}$, but cannot catalyze the methylation of the surrounding histones (*Diagouraga et al., 2018*). In B6-Tg(YF) mice, 5hmC was enriched at the B6 DMC1 sites (bound by PRDM9$^{Dom2}$), as expected, but not at the RJ2 DMC1 sites bound by PRDM9$^{Cst-YF}$ (*Figure 4E*). We conclude that PRDM9 binding is not sufficient and that its methyltransferase activity is also required for 5hmC enrichment. Then, to test whether 5hmC enrichment required also DSB activity (or downstream events), we analyzed 5hmC enrichment at hotspots in *Spo11* KO mice in which DSB formation is defective. In these mice, 5hmC levels were identical to wild-type mice (*Figure 4F*). This result indicates that DSB formation is not required for 5hmC, and that a step between PRDM9 histone modification and DSB formation leads to 5hmC enrichment at meiotic hotspots.

We then analyzed the correlation of 5hmC enrichment with the strength of PRDM9, H3K4me3, SPO11-oligos and DMC1 enrichment. SPO11-oligos are the molecular intermediates generated after DSB formation by endonucleolytic cleavage of the strand to which SPO11 is covalently bound (*Neale et al., 2005*). SPO11-oligos data are available only for the B6 genotype (*Lange et al., 2016*). DMC1 enrichment reflects DSB formation, but is also influenced by features of DSB repair, and is not directly proportional to SPO11-oligos (*Hinch et al., 2019*). The correlation plots revealed that in the RJ2 strain, 5hmC was best correlated with PRDM9 and H3K4me3 enrichment, and in the B6 strain, with SPO11-oligo enrichment (*Figure 4—figure supplement 1C*). In both strains, the weakest correlation was between 5hmC and DMC1. This suggests that 5hmC enrichment at hotspots is better correlated with events directly linked to PRDM9 binding and DSB activity, rather than to DSB repair, which is consistent with the functional dependency reported above.

As 5hmC level at hotspots may depend on the density of CpG dinucleotides and of 5mC, it was important to examine the same correlations in function of the CpG content within hotspots (*Figure 5—figure supplement 1A*). Over a +/- 250 bp window around hotspot centers, the mean number of CpG was 4.4 (0.88 CpG/100b) in B6, and 4.3 (0.86 CpG/100b) in RJ2. Of note, the consensus motif for PRDM9$^{Dom2}$ and PRDM9$^{Cst}$ does not include CpGs (*Baker et al., 2015*; *Grey et al., 2017*). The slight increase in CpG density around PRDM9$^{Dom2}$ hotspot was expected due to the process of

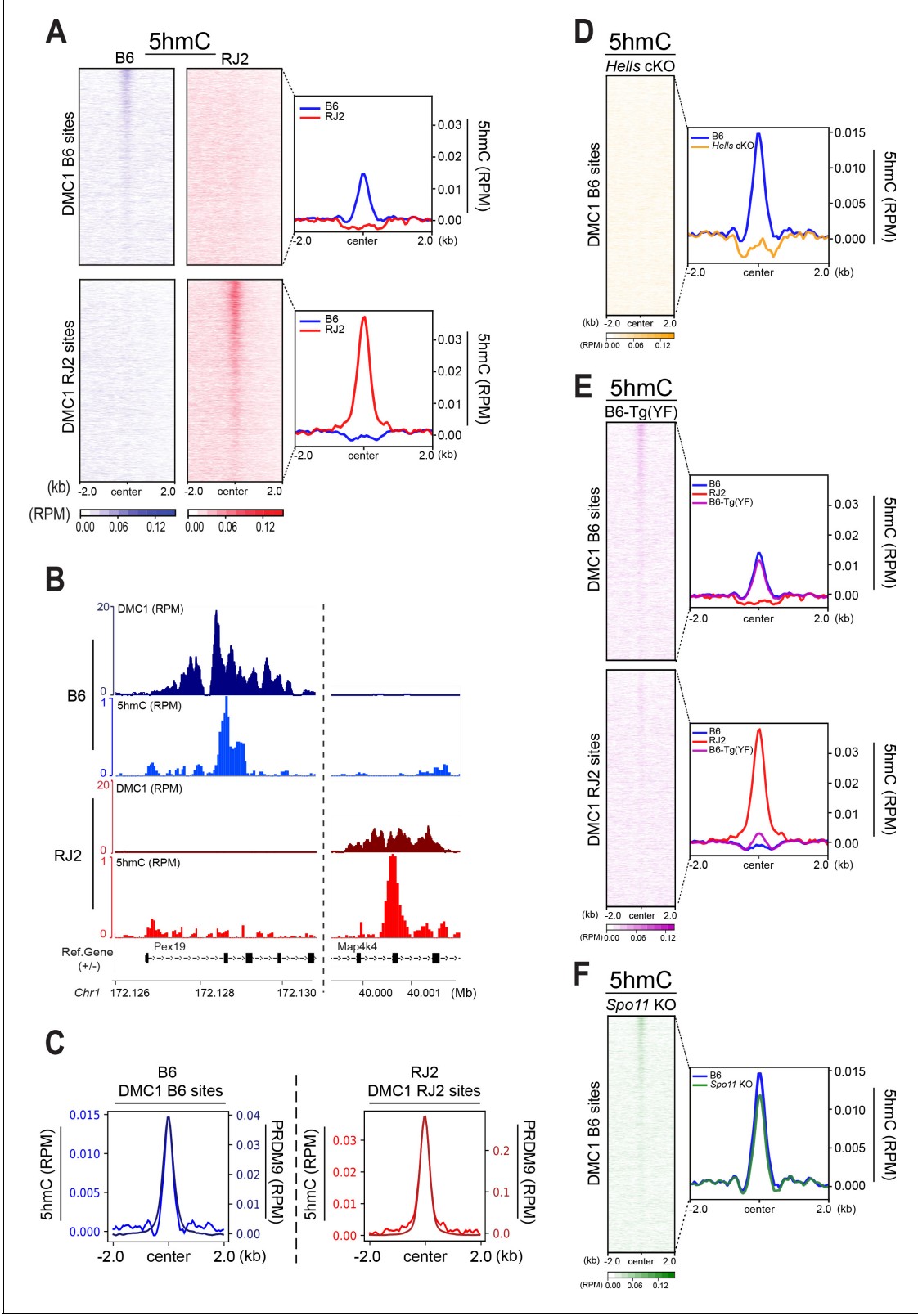

**Figure 4.** 5hmC is enriched at PRDM9-dependent sites and correlates with PRDM9 occupancy. (**A**) Average read enrichment and heatmaps showing 5hmC enrichment in the B6 (blue) and RJ2 (red) strains. Signal was calculated in a +/- 2 kb window around hotspot centers (determined by DMC1-SSDS). 5hmC enrichment was calculated from pooled replicates within 50bp bins and normalized by reads per million (RPM) and input. The sites on the heatmaps are ranked by decreasing DMC1-SSDS signal intensity from top to bottom. (**B**) Read distribution from DMC1 and 5hmC ChIP-seq

*Figure 4 continued on next page*

*Figure 4 continued*

experiments in the B6 (blue) and RJ2 (red) strains at representative DMC1 PRDM9$^{Dom2}$ (B6) and PRDM9$^{Cst}$ (RJ2) specific sites on chromosome 1. Read distribution was calculated from pooled replicates within 50bp bins and normalized by library size and input, except for the DMC1 ChIP experiments. (C) Average read enrichment showing 5hmC enrichment (left y axis) and PRDM9 read enrichment (right y axis) in the B6 (blue) and RJ2 (red) strains centered in a +/- 2 kb window around DMC1 B6 and DMC1 RJ2 sites, respectively. Read distribution was calculated from pooled replicates within 50bp bins and normalized by library size and input. (D) 5hmC signal at hotspots is HELLS-dependent. Average read enrichment showing 5hmC in the B6 (blue) and *Hells* cKO (orange) strains centered in a +/- 2 kb window around the hotspot centers (DMC1-SSDS B6 sites). 5hmC enrichment was calculated from pooled replicates within 50bp bins and normalized by read per million (RPM) and input. (E) 5hmC signal at hotspots is dependent on PRDM9 methyltransferase activity. Average read enrichment showing 5hmC in the B6 (blue), RJ2 (red) and B6-Tg(YF)(magenta) strains centered in a +/- 2 kb window around the hotspot centers (DMC1-SSDS B6 and RJ2 sites). 5hmC enrichment was calculated from pooled replicates within 50bp bins and normalized by read per million (RPM) and input. (F) 5hmC signal at hotspots is independent of DSB formation. Average read enrichment showing 5hmC enrichment in the B6 (blue) and *Spo11* KO (green) strain centered in a +/- 2 kb window around the hotspot centers (DMC1-SSDS B6 sites). 5hmC enrichment was calculated form pooled replicates within 50bp bins and normalized by read per million (RPM) and input. The duplicate analysis for all genotypes is shown in *Figure 4—figure supplement 2*.

The online version of this article includes the following figure supplement(s) for figure 4:

**Figure supplement 1.** Similar distributions of 5hmC and PRDM9 enrichments.

**Figure supplement 2.** Reproducibility of 5hmC enrichment.

---

GC-biased gene conversion (*Duret and Galtier, 2009*) that leads to higher GC content at meiotic recombination hotspots (*Clément and Arndt, 2013*; *Grey et al., 2017*). We then examined the methylation level at CpGs within +/- 250 bp from B6 and RJ2 hotspot centers that contained at least one CpG using published sodium bisulfite data from B6 samples (*Gaysinskaya et al., 2018*). As control, we analyzed the methylation level at four different types of genomic sites: (i) two families of transposable elements (LINE and IAP), and (ii) two sets of imprinted control regions (ICRs): one set methylated only in females (female-specific) and the other methylated only in males (male-specific). As shown before (*Ferguson-Smith, 2011*), we observed low cytosine methylation levels at female-specific ICRs and high methylation levels at male-specific ICRs and the transposable elements LINE and IAP (*Figure 5—figure supplement 1B*). Meiotic hotspots specific for each strain (B6 and RJ2) showed a similar median methylation level of at least 90% at all stages analyzed (B type spermatogonia, leptotene and pachytene spermatocytes), with a level comparable to what observed in the genome (*Figure 5—figure supplement 1B*). Note that B6 hotspots, but not RJ2 hotspots, were active in the strain where methylation was monitored. This suggests that overall, in the cell population, the level of hotspot methylation is high already before they are bound by PRDM9, with no further detectable local increase of cytosine modification upon PRDM9 binding. This property was also mentioned in a recent study where the methylation level at DSB sites was measured by Nucleosome Occupancy and Methylome sequencing (NOMe-seq) at different stages during spermatogenesis (*Chen et al., 2020*). As sodium bisulfite sequencing allows detecting both 5mC and 5hmC, we propose that the 5hmC enrichment we detected at active hotspots results from the conversion of pre-existing 5mC at these sites rather than de novo modification of unmodified cytosines.

We then evaluated the correlation between CpG content and 5hmC enrichment, by clustering hotspots according to their number of CpG dinucleotides within a region of +/- 250 bp around the center. We defined four groups of sites: (i) no CpG, (ii) 1–2 CpG, (iii) 3–5 CpG, and (iv)≥6 CpG dinucleotides. The average plots revealed that sites with higher numbers of CpGs tended to have higher 5hmC enrichment, in agreement with the fact that CpGs are the substrates for this modification (*Figure 5A–B*, *Figure 5—figure supplement 1C*). In contrast, the number of CpGs was not correlated with hotspot activity. This is shown by the overlapping curves of average plots for the four groups of CpG content of PRDM9, H3K4me3, and DMC1 enrichment (*Figure 5—figure supplement 1D*). Heatmaps within each group of hotspots with similar numbers of CpGs revealed also that for a given CpG content, the 5hmC level correlated with the PRDM9, H3K4me3, and DMC1 site intensity (*Figure 5—figure supplement 1D*), an observation coherent with the functional dependency on PRDM9 binding and methyltransferase activity reported above.

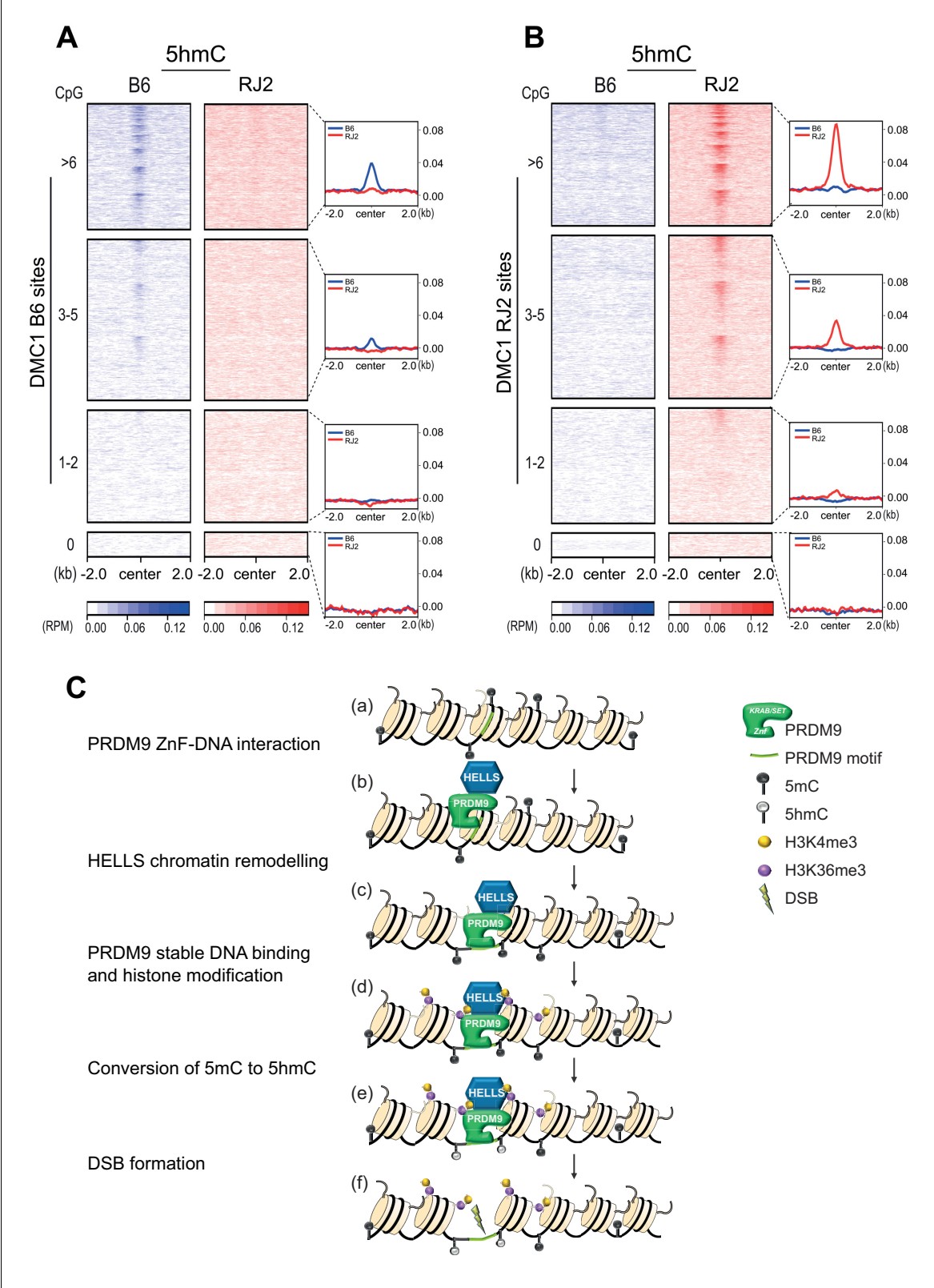

**Figure 5.** 5hmC enrichment at DSB sites sorted by CpG content and model. (**A**) Average read enrichment and heatmaps showing 5hmC enrichment at DMC1-SSDS B6 sites. (**B**) Average read enrichment and heatmaps showing 5hmC enrichment at DMC1-SSDS RJ2 sites. In both panels, the signal was calculated in the B6 (blue) and RJ2 (red) strain in a +/- 2 kb window around the hotspot centers (determined by DMC1-SSDS) and sorted by CpG content with hotspots clustered in four groups: hotspots without CpGs, and three groups of similar size containing increasing numbers of CpGs (1–2, 3–

*Figure 5 continued on next page*

*Figure 5 continued*

5, and at least 6 CpGs). The number of sites for each group is indicated in the Methods section. For a given CpG content, the sites are ranked by decreasing DMC1-SSDS signal intensity from top to bottom. CpG content was calculated in a +/- 250 bp window around the hotspot centers. The same analysis but at PRDM9 sites is shown in *Figure 5—figure supplement 1C*. (C) Model for the targeting of DSB activity by PRDM9/HELLS in mouse male meiosis. (a) A potential PRDM9-binding site is a specific DNA motif in a region of chromatin with no specific feature. For each nucleosome, only two histone tails (H3) are shown. (b) The zinc-finger domain of PRDM9 (ZnF) interacts with specific DNA motifs. PRDM9 may be interacting as a complex with HELLS before binding to its target sites as suggested by *Spruce et al., 2020*. (c) HELLS promotes chromatin remodeling, enhancing accessibility of PRDM9 to its DNA motif and a stable interaction. (d) PRDM9 methyltransferase catalyzes H3K4me3 and H3K36me3 on adjacent nucleosomes. These histone modifications may or not be symmetric (*Lange et al., 2016*). H3K9Ac is also known to be enriched near PRDM9 -binding site at this stage (not shown). (e) Putative methylated cytosines (5mC) near the PRDM9-binding site are converted to 5hmC, suggesting the recruitment of a TET enzyme. (f) DSB forms at or adjacent to the PRDM9-binding site.

The online version of this article includes the following figure supplement(s) for figure 5:

**Figure supplement 1.** CpG content and 5hmC at meiotic hotspots.

## Discussion

### A chromatin remodeler for PRDM9 binding

In 2010, the discovery that PRDM9 is the factor directing the location of meiotic DSBs in mammals raised several questions about the underlying molecular process. One of them was how the zinc- finger domain of PRDM9 gains access to its DNA-binding motifs. These DNA sites have no other reported function than being bound by PRDM9, and this occurs specifically in meiotic cells where *Prdm9* is expressed. These sites are located throughout the genome, in genic and intergenic regions, and they do not appear to have any intrinsic feature beyond their DNA sequence. The only identified landmark is their higher GC content relative to their flanking regions that appears progressively over generations due to the process of GC-biased gene conversion during meiotic DSB repair at these sites (*Clément and Arndt, 2013*; *Grey et al., 2017*). Moreover, PRDM9-binding sites can differ between mouse strains because the PRDM9 DNA-binding domain is highly mutable, and multiple variants with distinct DNA-binding specificity are present in *Mus musculus* (*Buard et al., 2014*; *Kono et al., 2014*; *Vara et al., 2019*).

One of the major advances of this study, together with the parallel study from C. Baker's group (*Spruce et al., 2020*), is the identification of the interaction between PRDM9 and HELLS, and the evidence of its essential role in allowing PRDM9 to access and stably bind to its binding sites (*Figure 5C*). The interaction between PRDM9 and HELLS was detected by IP-mass spectrometry and by yeast two-hybrid assays (this study) and by co-IPs (*Spruce et al., 2020*). In addition, and consistent with these interactions, ChIP experiments showed that HELLS is enriched at least at a fraction of PRDM9 sites, presumably the ones with the most efficient PRDM9 binding (*Spruce et al., 2020*). The chromatin configuration of PRDM9-binding sites has been analyzed by accessibility to MNase and to the transposase Tn5 (ATAC-seq). Before PRDM9 expression (in spermatogonia), most PRDM9-binding sites do not reveal specific accessibility compared with flanking genomic sequences (*Chen et al., 2020*). Conversely, when PRDM9 is expressed (leptonema), the binding sites show increased chromatin accessibility along few hundred base pairs on both sides of the PRDM9-binding site (*Baker et al., 2014*; *Yamada et al., 2020*). The footprint of PRDM9 binding is not detected by ATAC-seq, suggesting a short residency time (*Spruce et al., 2020*). As PRDM9 promotes H3K4me3 and H3K36me3 deposition on flanking nucleosomes, the chromatin organization at these sites can be examined and reveals a well-positioned arrangement of modified nucleosomes around PRDM9-binding sites. Overall, a striking change of chromatin configuration is observed upon PRDM9 binding, and HELLS plays an essential role in this process. This conclusion is based on the observation that in *Hells* cKO spermatocytes, PRDM9 binding (this study), PRDM9 dependent-H3K4me3 deposition, and chromatin accessibility (*Spruce et al., 2020*) cannot be detected at PRDM9-binding sites.

HELLS belongs to the SNF2-like family of chromatin remodelers (*Flaus et al., 2006*), but unlike other members of this family no ATP-dependent nucleosome remodeling activity could be detected in vitro (*Burrage et al., 2012*). However, in *Xenopus laevis* extracts, HELLS promotes nucleosome remodeling when forming a complex with CDCA7 that contains a 4-CXXC zinc-finger domain (*Jenness et al., 2018*). Therefore, it could be anticipated that in meiotic cells, HELLS is brought to PRDM9-binding sites by interacting with PRDM9. HELLS-dependent steps, such as nucleosome

repositioning and/or stabilization and the subsequent opening of chromatin, could further stabilize the interaction of the PRDM9 zinc-finger domain with these sites, in a process partly similar to the one described for pioneer transcription factors (*Mayran and Drouin, 2018*). Functionally, HELLS absence leads to a phenotype comparable to the one observed upon *Prdm9* genetic ablation. Indeed, in *Hells* cKO mice, meiotic DSB activity is undetectable at PRDM9 sites and is redirected to sites of open chromatin, such as promoters and enhancers (this study; *Spruce et al., 2020*), which are called default sites, like in *Prdm9* KO mice (*Brick et al., 2012*). Therefore, HELLS is an essential determinant of meiotic DSB localization in mice.

Could HELLS have additional role(s) beyond promoting PRDM9 binding? As observed in *Prdm9* KO mice, *Hells* cKO mice show a partial defect in DSB repair and homologous synapsis. By promoting nucleosome reorganization at PRDM9-binding sites, HELLS may contribute to DNA repair. This contribution may concern the chromatin of the broken and also the uncut chromatid because it has been proposed that PRDM9 binds not only to the chromatid where DSBs will occur but also to the intact template (*Davies et al., 2016*; *Hinch et al., 2019*). Such 'symmetric binding' might enhance interhomolog repair. In theory, HELLS activity could participate in DSB repair by regulating chromatin organization on the broken chromatid (for instance for strand resection), and on the uncut chromatid (for strand invasion). In support of this hypothesis, it has been shown that in somatic cells, HELLS is involved in and facilitates DSB repair (*Burrage et al., 2012*; *Kollárovič et al., 2020*). In somatic cells, HELLS might facilitate end-resection by interacting with and recruiting C-terminal interacting protein (CtIP) (*Kollárovič et al., 2020*). HELLS implication in genome integrity has also been detected in *Neurospora crassa* (*Basenko et al., 2016*) and *Saccharomyces cerevisiae* (*Litwin et al., 2017*). In *S. cerevisiae*, the HELLS orthologue Irc5 is required for DNA damage tolerance, and this function implies the loading of the cohesin complex at replication forks (*Litwin et al., 2018*). In *S. cerevisiae*, cohesin recruitment facilitates DSB repair (*Ström et al., 2004*; *Unal et al., 2004*). The hypothesis of a role for HELLS in meiotic DSB repair through enhancing end-resection and/or cohesin loading remains to be tested.

In other cellular contexts, HELLS is a major regulator of DNA methylation, specifically for the silencing of repeated DNA elements, and through the recruitment of the DNA methyltransferase DNMT3B (*Myant and Stancheva, 2008*; *Zhu et al., 2006*). This has an impact on DNA methylation genome-wide (*Ren et al., 2019*; *Yu et al., 2014a*). Besides altering epigenetic features and the expression of transposable elements (TE), HELLS absence in mouse tissues (brain and liver) and in fibroblasts has very limited consequences on gene expression (*Huang et al., 2004*; *Yu et al., 2014b*). In *Hells* KO mouse oocytes, the DNA methylation level of some TE families is reduced and their expression is increased; however, the expression of several meiotic genes is not affected (*De La Fuente et al., 2006*). The consequences of *Hells* deficiency on DNA methylation and expression have not been tested in spermatocytes. However, in the absence of HELLS, major epigenomic alterations in non-repeated DNA are not expected during meiosis, and consistently, H3K4me3 level at promoters is not altered in *Hells* cKO spermatocytes (*Spruce et al., 2020*).

## The implication of 5-hydroxymethylcytosine at meiotic DSB sites

The presence of 5hmC at DSB hotspots was first reported by Brick and colleagues (*Brick et al., 2018*) using genome-wide data on cytosine methylation and hydroxyl-methylation patterns in mouse spermatocytes, mainly at the pachytene stage (*Hammoud et al., 2014*). Here, we found that this DNA modification is also present at hotspots earlier in meiosis, at leptotene-zygotene stages, when DSB formation takes place. By assessing 5hmC in mouse strains that carry different *Prdm9* alleles (*Prdm9$^{Dom2}$* and *Prdm9$^{Cst}$*), we found that 5hmC deposition depends on PRDM9 DNA-binding specificity. Moreover, we detected 5hmC enrichment in a narrow window of about +/- 250bp around the center of PRDM9-binding sites. Remarkably, PRDM9 binding is not sufficient and PRDM9 methyltransferase activity also is required for 5hmC enrichment. As we showed that 5hmC presence at hotspots does not require SPO11, we propose that 5hmC is promoted by a PRDM9-dependent chromatin modification step before DSB formation (*Figure 5C*). Therefore, 5hmC is a new feature of the local signature of active hotspots, like the histone modifications H3K4me3, H3K4me36 and H3K9ac (*Buard et al., 2009*; *Davies et al., 2016*; *Grey et al., 2017*; *Lam et al., 2019*; *Powers et al., 2016*).

The next question concerns the mechanism of the 5hmC enrichment at PRDM9-dependent hotspots. This enrichment is not observed in somatic tissues (*Brick et al., 2018*), which is consistent

with the PRDM9 dependency we observed and with the PRDM9-specific expression at the leptotene stage of the meiotic prophase (*Jung et al., 2019*; *Spruce et al., 2020*). In the mouse male germline, a high level of DNA methylation is induced genome-wide during germline development in spermatogonia before meiosis entry and is maintained during meiotic prophase with a transient reduction at preleptonema (*Gaysinskaya et al., 2018*). A similar high DNA methylation level is observed at meiotic DSB sites (*Figure 5—figure supplement 1D*; *Chen et al., 2020*). Therefore, it could be hypothesized that a TET enzyme promoting the conversion of 5mC to 5hmC (*Ito et al., 2010*) is recruited upon or concomitantly with PRDM9 binding to its sites. One possible scenario could be that TET recruitment involves HELLS. Indeed, HELLS can interact with one, two, or all three TET enzymes, depending on the cell type (MCF-7 cells, HEK293T, mouse embryonic stem cells) (*de Dieuleveult et al., 2020*; *Jia et al., 2017*), and co-localize with 5hmC when stably expressed in HK1 cells (*Jia et al., 2017*). As no evidence of HELLS/TET interaction in meiotic cells is available, a PRDM9-dependent chromatin modification might be implicated in recruiting the putative TET activity. ZCWPW1, a reader of H3K4me3 and H3K36me3 that is required for efficient DSB repair (*Huang et al., 2020*; *Mahgoub et al., 2020*; *Wells et al., 2019*), might be directly or indirectly involved in this recruitment.

The function of 5hmC at meiotic hotspots is unknown. At least two non-exclusive consequences of 5hmC can be envisioned. First, as 5hmC has been associated with sites of open chromatin, such as active and poised enhancers, in several cell types (*Sérandour et al., 2012*; *Stroud et al., 2013*; *Szulwach et al., 2011*), it could have an active role in recruiting partners or stabilizing interactions, similarly to the recruitment of factors described in neuronal progenitor cells (*Spruijt et al., 2013*). Second, it has been shown that 5hmC prevents the binding of several methyl-CpG-binding proteins (*Jin et al., 2010*). One or both of these consequences of the conversion of 5mC to 5hmC could have a positive effect on DSB repair at meiotic hotspots in male meiosis. We favor the second scenario, in which the conversion of 5mC to 5hmC allows antagonizing the binding of factors with affinity for 5mC and which could interfere with meiotic recombination. Indeed, it is difficult to reconcile a positive role for 5hmC with the observation that DSB formation and repair is efficient at hotspots without CpGs in spermatocytes, and at all hotspots in oocytes, which have a low global level of cytosine methylation (*Seisenberger et al., 2012*).

The control of initiation sites of meiotic recombination by PRDM9 underlies a sophisticated regulation that goes beyond the simple binding to specific DNA motifs in the genome. Clearly, our findings and those from Baker's laboratory (*Spruce et al., 2020*) indicate that the control of chromatin is an important step for DSB formation and repair. PRDM9 and HELLS drive epigenetic modifications before and independently of DSB formation, setting the stage for downstream steps. Not only histone modifications but also DNA methylation appears to be a potential additional level of regulation of meiotic recombination, with potential distinct consequences during male and female meiosis where some differences in hotspot activity have been detected (*Brick et al., 2018*) and from the analysis of 5hmC in the male germ line presented in this study. These observations also highlight the need of understanding the sex-specific features of meiotic recombination in general.

# Materials and methods

## Key resources table

| Reagent type (species) or resource | Designation | Source or reference | Identifiers | Additional information |
|---|---|---|---|---|
| Mouse strains | | | | |
| | C57BL/6JOlaHsd | Envigo | C57BL/6JOlaHsd | Named B6 |
| | B10.MOLSGR(A)-(D17Mit58-D17Jcs11)/Bdm (RJ2) | *Grey et al., 2009* | MGI:5319075 | Named RJ2 |
| | B6;129P2 < Prdm9tm1Ymat>/J | *Hayashi et al., 2005* | MGI:3624989 | Named *Prdm9* KO |

*Continued on next page*

*Continued*

| Reagent type (species) or resource | Designation | Source or reference | Identifiers | Additional information |
|---|---|---|---|---|
| | Spo11 < tm1Mjn> | *Baudat et al., 2000* | MGI:2178805 | Named *Spo11* KO |
| | Hells < tm1a (EUCOMM) Wtsi/Ieg> | EUCOM *Bradley et al., 2012* | MGI:4431905 | |
| | C57BL/6 Tg(CAG-Flpo)1Afst | *Kranz et al., 2010* | MGI:4453967 | |
| | C57BL/6 Tg(CMV-cre)1Cgn | *Schwenk et al., 1995* | MGI:2176180 | |
| | Tg(Stra8-icre) 1Reb/J <(Stra8-iCre)> | *Sadate-Ngatchou et al., 2008* | MGI:3779079 | |
| | Tg(RP23-159N6*)23Bdm | *Diagouraga et al., 2018* | MGI:5565212 | Named B6-Tg(YF) |
| Cell lines | | | | |
| | HeLa | ATCC | HeLa S3 ATCC CCL-2.2 | |
| Yeast strains | | | | |
| | AH109 | *James et al., 1996* | | *S. cerevisiae* |
| | Y187 | *Harper et al., 1993* | | *S. cerevisiae* |
| Recombinant DNA reagents | | | | |
| | PRDM9A-Flag-HA-Nt into retroviral pOZ-FH-N vector | This study | N/A | Vector from Addgene DB3781 |
| | PRDM9A-Flag-HA-Ct into retroviral pOZ-FH-C vector | This study | N/A | Vector from Addgene cat# 32516 |
| | pGAD GH for fusion to Gal4 activation domain, modified for Gateway cloning | *Van Aelst et al., 1993* | Clontech No. 638853 | LEU2 marker |
| | pAS2dd for fusion to Gal4 DNA-binding domain, modified for Gateway cloning | *Fromont-Racine et al., 1997* | | TRP1 marker |
| | pB29 for PRDM9 (aa 1–511) expression fused to LexA for yeast two-hybrid screen | Hybrigenics | | |
| Antibodies | | | | |
| | Guinea-pig anti-SYCP3 | *Grey et al., 2009* | N/A | Home-made WB: 1/2000 IF: 1/500 |
| | Rabbit anti-SYCP1 | Abcam | Cat# ab15090 RRID:AB_301636 | IF: 1/400 |
| | Rabbit anti-DMC1 | Santa Cruz | Cat# scH100 RRID:AB_2277191 | IF: 1/200 |
| | Goat anti-DMC1 | Santa Cruz | Cat# scC20 RRID:AB_2091206 | ChIP: 24 μg |

*Continued*

| Reagent type (species) or resource | Designation | Source or reference | Identifiers | Additional information |
|---|---|---|---|---|
| | Rabbit anti-HELLS | Novus | Cat# NB 100–278 RRID:AB_350198 | WB: 1/2000 IF: 1/200 |
| | Mouse monoclonal anti-HELLS | Santa Cruz | Cat# sc46665 RRID:AB_627895 | IF: 1/100 |
| | Mouse monoclonal anti-phospho-histone H2AFX (Ser139) | Millipore | Cat# MP05-636 RRID:AB_309864 | Named γH2AFX IF: 1/10000 |
| | Rabbit anti-Gal4 activation domain (GAD) (Millipore, 06–283) | | Now at Sigma-Aldrich Cat# ABE476 | WB: 1/3000 |
| | Rabbit anti-Gal4 DNA-Binding domain | Sigma–Aldrich | Cat# G3042 RRID:AB_439688 | WB: 1/2000 |
| | Rat monoclonal anti-Tubulin [YOL1/34] | Abcam | Cat# ab 6161 RRID:AB_305329 | WB: 1/3000 |
| | Rabbit anti-5hmC | Active Motif | Cat# AM 39791 RRID:AB_2630381 | hMeDIP: 5 μg |
| | Rabbit anti-PRDM9 | *Grey et al., 2017* | N/A | Home-made WB: 1/2000 IF: 1/200 ChIP: 4 μg |
| | Rabbit anti-H3K4me3 | Abcam | ab8580 RRID:AB_306649 | ChIP: 4 μg |
| | Goat anti-rabbit IgG-HRP | Pierce | Cat# 1858415 RRID:AB_1185567 | WB: 1/10000 |
| | Goat anti-Guinea-pig IgG-HRP | Jackson Immuno Research | Cat# 706-035-148 RRID:AB_2340447 | WB: 1/3000 |
| | Goat anti-rabbit IgG-Alexa 555 | Thermo Fisher Scientific | Cat# ab150078 RRID:AB_2535849 | IF: 1/400 |
| | Goat anti-guinea-pig IgG-Alexa 488 | Thermo Fisher Scientific | Cat# ab150185 RRID:AB_2534117 | IF: 1/400 |
| | Donkey anti-mouse IgG-Alexa 680 | Thermo Fisher Scientific | Cat# ab175774 RRID:AB_2534014 | IF: 1/100 |
| | Donkey anti-mouse IgG-Alexa 647 | Thermo Fisher Scientific | Cat# ab150107 RRID:AB_162542 | IF: 1/400 |
| Oligonucleotides | | | | |
| | Genotyping *Hells* cKO mice, see *Supplementary file 2* | This study | | |
| | RT-qPCR, see *Supplementary file 3* | This study, *Buard et al., 2009*, *Diagouraga et al., 2018* | N/A | |
| Commercial assays or kits | | | | |

*Continued on next page*

*Continued*

| Reagent type (species) or resource | Designation | Source or reference | Identifiers | Additional information |
|---|---|---|---|---|
| | DeadEnd Fluorometric TUNEL System | Promega | Cat# G3250 | |
| | Anti-HA beads | Santa Cruz | Cat# sc-500773 | |
| | EZview anti-FLAG M2 Affinity Gel | Sigma–Aldrich | Cat# F2426 | |
| | hMeDIP Kit | Actif Motif | Cat# AM55010 | |
| | NEB Next Ultra Library Preparation Kit | New England Biolabs | Cat# NEB7370S | |
| | ChIP-IT High Sensitivity Kit | Actif Motif | Cat# AM53040 | |
| | MMLV-based retroviral transduction system | *Nakatani and Ogryzko, 2003* | N/A | |
| Chemical compounds | | | | |
| | HA peptide | Covance | Cat #PEP-101P-1000 | |
| | FLAG peptide | Sigma | Cat #F4799 | |
| | Optiprep Idoixanol | Sigma–Aldrich | Cat# D1556 | |
| | Sytox Green | Thermo Fisher Scientific | Cat# S70020 | |
| | WIN 18466 | Tocris Bioscience | Cat# 4736 | *Hogarth et al., 2013* |
| | Retinoic Acid | Sigma–Aldrich | Cat# R2625 | |
| Deposited data | | | | |
| | Mass spectrometry proteomics | ProteomeXchange Consortium | Dataset identifier PXD017337 | |
| | NGS SSDS ChIP (DMC1) and hMeDIP | GEO | GSE145768 | |
| Softwares and Algorithms | | | | |
| | Bowtie 2 | | http://bowtie-bio.sourceforge.net/bowtie2/index.shtml | |
| | Modified BWA algorithm | *Khil et al., 2012* | N/A | |
| | Tim Galore! | | https://www.bioinformatics.babraham.ac.uk/projects/trim_galore/ | |
| | Bismark | | https://www.bioinformatics.babraham.ac.uk/projects/bismark/ | |

*Continued on next page*

*Continued*

| Reagent type (species) or resource | Designation | Source or reference | Identifiers | Additional information |
|---|---|---|---|---|
| | Bedtools suite | | https://bedtools.readthedocs.io/en/latest/content/bedtools-suite.html | |

## Mouse strains

The following mouse strains were used: C57BL/6JOlaHsd (hereafter B6), B10.MOLSGR(A)-(D17Mit58-D17Jcs11)/Bdm (RJ2) (*Grey et al., 2009*), B6;129P2-*Prdm9*$^{tm1Ymat}$/J (B6 PRDM9$^{KO}$) (*Hayashi et al., 2005*), *Spo11*$^{tm1Mjn}$ (B6 SPO11$^{KO}$) (*Baudat et al., 2000*), C57BL/6J-Tg(RP23-159N6*) 23Bdm (B6-Tg(YF)) (*Diagouraga et al., 2018*). *Hells*$^{tm1a(EUCOMM)Wtsi/Ieg}$ mice EUCOM consortium *Bradley et al., 2012* have a C57BL/6N genetic background with the *Prdm9*$^{Dom2}$ allele. These mice were mated with mice expressing FLP from the CMV promoter (C57BL/6 Tg(CAG-Flpo)1Afst) (*Kranz et al., 2010*) to generate a floxed (*Hells*$^{fl}$) allele. *Hells*$^{fl/fl}$ mice were mated with mice that express CRE under the control of the CMV promoter (C57BL/6 Tg(CMV-cre)1Cgn) (*Schwenk et al., 1995*) to generate *Hells*-deleted heterozygous mice (*Hells*$^{+/-}$). *Hells*$^{+/-}$ mice were mated with Tg (Stra8-icre)1Reb/J (*Stra8-Cre*$^{Tg}$) mice (*Sadate-Ngatchou et al., 2008*) to generate *Hells*$^{+/-}$;*Stra8-Cre*$^{Tg}$ mice. By crossing *Hells*$^{fl/fl}$ mice with *Hells*$^{+/-}$;*Stra8-Cre*$^{Tg}$ mice, *Hells*$^{fl/-}$;*Stra8-Cre*$^{Tg}$ (*Hells* cKO) mice and *Hells*$^{fl/+}$, *Hells*$^{fl/+}$ *Stra8-Cre*$^{Tg}$ or *Hells*$^{fl/-}$ (*Hells* CTRL) mice were obtained. RJ2 mice have a C57BL/10 genetic background, very similar to that of B6, and carry the *Prdm9*$^{Cst}$ allele. B6-Tg(YF) mice carry both the endogenous wild-type *Prdm9*$^{Dom2}$ allele and the transgenic methyltransferase-dead *Prdm9*$^{Cst-YF}$ allele (Y357F mutation on *Prdm9*$^{Cst}$ allele) on a BAC transgene. All experiments were carried out according to the CNRS guidelines and were approved by the ethics committee on live animals (project CE-LR-0812 and 1295).

## HeLa cells

### Generation of HeLa cells that express human PRDM9$^A$ tagged with Flag-HA

To generate HeLa S3 cells that express PRDM9 tagged with Flag and HA, the previously described MMLV-based retroviral transduction system was used (*Nakatani and Ogryzko, 2003*). Human PRDM9$^A$ was cloned in the pOZ-FH-N and pOZ-FH-C derivative vectors to express PRDM9$^A$-Flag-HA-Nt and PRDM9$^A$-Flag-HA-Ct, respectively. The HeLa S3 cell lines expressing PRDM9$^A$-Flag-HA-Nt and PRDM9$^A$-Flag-HA-Ct were generated. HeLa S3 cells were authenticated by STR profiling through Eurofins. Cells were regularly tested with MycoAlert mycoplasma detection kit (Lonza, LT07-218), the ratio of ATP level before and after the addition of the MycoAlert reagent was below 0.9, indicating that the HeLa cells used for these experiments were mycoplasma free.

### Preparation of HeLa cell protein extracts

Nuclear protein extracts were prepared from 1 L (~10$^8$ cells) of cell culture using the Dignam protocol (*Dignam et al., 1983*) with minor modifications. Extracts were prepared from cells that express PRDM9$^A$-Flag-HA-Nt, PRDM9$^A$-Flag-HA-Ct or without expression vector.

### Immunoprecipitation of HeLa cell protein extracts

The PRDM9 complex was purified by immunoprecipitation (IP) using anti-FLAG (IP-FLAG) and -HA antibodies (IP-HA). About 35 mg of proteins from each nuclear fraction were used. FLAG affinity purification was performed with EZview anti-FLAG M2 Affinity Gel (Sigma). Elution was performed with 0.2 mg/ml of FLAG peptide. HA affinity purification was performed with anti-HA beads (Santa Cruz). Elution was performed with 0.4 mg/ml HA peptide (eluate 1 and 2) and 2 mg/ml HA peptide (eluate 3). Eluates 1 and 2 were analyzed on 4–15% acrylamide gels by silver staining (Silver Quest Staining Kit, Invitrogen).

## Mass spectrometry of HeLa cell immunoprecipitates

Eluates 1 and 2 of IP-HA were pooled and analyzed by mass spectrometry. The pooled proteins were precipitated with the TCA method using the ProteoExtract Protein Precipitation Kit (Calbiochem). All samples purified from protein extracts of PRDM9-Nt- and -Ct-expressing, or non-PRDM9-expressing HeLa cells were analyzed using a Velos-Orbitrap Pro mass spectrometer (Thermo Scientific) at the Taplin Mass Spectrometry Facility. The mass spectrometry data were analyzed with GFY, an application developed in Gygi's laboratory (Harvard University). Pilot experiments were performed with size separation by gel electrophoresis and protein extraction from slices corresponding to the MW of 70–80 kD and of 95–120 kD before mass spectrometry (130927, samples 43346 to 43351). For the full proteomic analysis, whole samples were sequenced (131026, samples 43738 to 43740). The list of proteins is in *Supplementary file 1*. Proteins defined as contaminants according to the Crapome and Mitocheck databases (www.crapome.org and www.mitocheck.org/) were removed.

## Preparation of mouse testis protein extracts

For mass spectrometry experiments, nuclear extracts were prepared from mouse testes from 12 to 13 dpp B6 mice (n = 18). Proteins were extracted from nuclei following the Dignam protocol (*Dignam et al., 1983*).

For analysis of PRDM9 and HELLS expression during mouse spermatogenesis, whole cell extracts were prepared from frozen testes collected from 4, 6, 9, 12, 15 dpp, and adult RJ2 males. Extraction was performed by homogenizing cells with a Dounce homogenizer in 400 mM NaCl, 50 mM Hepes, 1% Triton X-100, 4 mM DTT, complete protease inhibitor, followed by sonication and centrifugation to remove debris.

For PRDM9 and HELLS expression analysis in testes from 22 dpp *Hells* CTRL and *Hells* cKO mice, nuclear extracts were prepared. Testes were homogenized in hypotonic buffer (10 mM Hepes, pH 8.0, 320 mM sucrose, 1 mM PMSF, 1x Complete protease inhibitor cocktail EDTA-free (Roche, Cat. Number 11873580001)) in a Dounce homogenizer. After centrifugation (1000xg at 4°C for 10 min), supernatants were collected and used as cytoplasmic fractions. Nuclear fractions were from pellets that were resuspended in RIPA buffer (50 mM Tris–HCl, pH 7.5, 150 mM NaCl, 1 mM EDTA, 1% NP-40, 0.5% Na-deoxycholate, 0.1% SDS, 1x Complete protease inhibitor EDTA-free (Roche)), sonicated and centrifuged to remove debris.

## Western blotting

For PRDM9 and HELLS expression analysis in testes from 22 dpp *Hells* CTRL and *Hells* cKO mice, nuclear fractions (40 µg) were analyzed by western blotting with affinity-purified rabbit anti-PRDM9 (1/2,000) (*Grey et al., 2017*) and rabbit anti-HELLS (NB100-278, Novus) (1/2,000) antibodies and Guinea-pig serum raised against the mouse SYCP3 residues 24–44 (1/2,000). Secondary antibodies were goat anti-rabbit IgG-HRP (1/10,000) (1858415, Pierce) and goat anti-Guinea-pig IgG-HRP (1/3,000) (706-035-148, Jackson Immuno Research).

For PRDM9 and HELLS expression during mouse spermatogenesis, 50 µg of whole cell extracts were analyzed by western blotting with affinity-purified rabbit anti-PRDM9 (*Grey et al., 2017*), mouse anti-HELLS (SC-46665, Santa Cruz) and rat anti-tubulin (ab6161, Abcam) antibodies.

## Immunoprecipitation of mouse testis nuclear protein extracts

IP-PRDM9 and IP-Control (mock) were performed with 4 µg of anti-PRDM9 antibody (*Grey et al., 2017*) or normal rabbit serum and 3.6–3.8 mg of nuclear proteins after pre-clearing with protein A or G Dynabeads (Invitrogen).

## Mass spectrometry of mouse testis protein samples

IP samples were analyzed after separation on 7.5% acrylamide gels and silver staining (Silver QuestTM Staining Kit, Invitrogen). Protein extraction and purification were monitored by western blotting with an anti-PRDM9 antibody (*Grey et al., 2017*). IP samples were analyzed on an LTQ Velos Pro Orbitrap Elite mass spectrometer (Thermo Scientific), and the obtained data were processed with the MaxQuant software at the Functional Proteomics Platform (IGF, Montpellier). The data outputs include the intensity-based absolute quantification (iBAQ) and label-free quantification

(LFQ) intensities for each protein. The iBAQ value is the sum of the intensities of all tryptic peptides for each protein. Therefore, iBAQ values are proportional to the protein molar quantities. LFQ intensities are based on the intensities of each protein and are normalized at multiple levels to ensure that the LFQ intensity profiles across samples accurately reflect the protein relative amounts. Raw data are available at Proteome Exchange. Samples are: 150310_MS_ver3, Res_PRMD9_150805 (two duplicates). The protein list with the quantifications is in *Supplementary file 1*. Data have been deposited in ProteomeXchange, reference PXD017337.

## Yeast two-hybrid assays

All plasmids used in yeast two-hybrid assays were cloned with the Gateway Gene Cloning Technology (Invitrogen) and transformed in the AH109 and Y187 haploid strains. These strains were transformed with Gal4 DNA-binding domain (GBD) and Gal4 activation domain (GAD) fusion plasmids, respectively. Purified colonies of diploid strains were streaked on SD media plates lacking leucine and tryptophan (LW), or leucine, tryptophan and histidine (LWH), or leucine, tryptophan and histidine with 5 mM amino-triazole (LWH+5mMAT). Interactions between GAD- and GBD-fusion proteins were evaluated after cell growth at 30°C for 3 days. For verification of protein expression, protein extracts were prepared and analyzed by western blotting, as previously described (*Imai et al., 2017*). The yeast two-hybrid screen was performed by Hybrigenics using a mouse cDNA library prepared using RNA from testes of 14–16 dpp mice.

## Antibodies

Guinea pig anti-SYCP3 (*Grey et al., 2009*), rabbit anti-SYCP1 (Abcam, 15090), rabbit anti-DMC1 (Santa Cruz, H100), rabbit anti-HELLS (Novus, NB100-278), mouse monoclonal anti-HELLS (Santa Cruz, sc46665), and mouse monoclonal anti-phospho-histone H2AFX (Ser139) antibody (γH2AFX) (Millipore, 05–636) were used for immunostaining. For IP, a home-made anti-PRDM9 antibody was used (*Grey et al., 2017*). For western blots, anti-PRDM9 (*Grey et al., 2017*), anti-HELLS (Novus, NB100-278), anti-SYCP3 (*Grey et al., 2009*), anti-GAD (Millipore, 06–283), anti-GBD (Sigma, G3042), and anti-tubulin (Abcam, ab6161) antibodies were used. For 5hmC analysis, a rabbit anti-5hmC antibody (Active Motif, 39791) was used. For DMC1 ChIP-SSDS, a goat anti-DMC1 antibody (Santa Cruz, C-20) was used. For conventional ChIP experiments, rabbit anti-PRDM9 (*Grey et al., 2017*) and rabbit anti-H3K4me3 (Abcam, ab8580) antibodies were used.

## Histological analysis of paraffin sections and TUNEL assay

Mouse testes were fixed in Bouin's solution for periodic acid schiff (PAS) staining, or in 4% paraformaldehyde/1X PBS for immunostaining or TUNEL assay, at room temperature overnight. Testes were then embedded in paraffin and cut in 3μm-thick slices. PAS-stained sections were scanned using the automated tissue slide-scanning tool of a Hamamatsu NanoZoomer Digital Pathology system. TUNEL assay was performed with the DeadEnd Fluorometric TUNEL System (Promega), according to the manufacturer's protocol.

## Immunostaining of nuclei spreads and fixed nuclei

Characterization of *Hells* cKO spermatocytes and meiotic staging of spermatocytes after synchronization were performed on nuclei spreads. Meiotic staging after Fluorescence-Activated Cell Sorting (FACS) was performed using fixed nuclei deposited on poly-lysine coated slides. Spreads were prepared with the dry down technique, as described (*Peters et al., 1997*), and immunostaining was performed as described (*Grey et al., 2009*). Staging criteria were the following: pre-leptotene nuclei had weak SYCP3 nuclear signal and no or very weak γH2AFX signal; leptotene nuclei were γH2AFX-positive and SYCP1-negative; early/mid zygotene nuclei had less than nine fully synapsed chromosomes; late zygotene had nine or more fully synapsed chromosomes; and pachytene cells had all chromosomes fully synapsed, excepted for the sex chromosomes. The following antibodies were used: rabbit anti-PRDM9 (*Grey et al., 2017*), (1:200), rabbit anti-HELLS (NB100-278, Novus, 1:200), mouse anti-HELLS (sc46665, Santa Cruz, 1:100), rabbit anti-DMC1 (H-100, Santa Cruz, 1:200), guinea-pig anti-SYCP3 (*Grey et al., 2009*, 1:500), anti-SYCP1 (ab15090, Abcam, 1:400) and anti-γ H2AFX (05–636, Millipore, 1:10,000).

## Synchronization of meiosis in male mice

The first wave of spermatogonia entry into meiosis initiates at 8 days postpartum (8 dpp). Then, spermatocytes progress to meiotic prophase and reach the leptotene, zygotene and pachytene stages at approximately 11, 13 and 15 dpp, respectively. Hence, the proportion of cells at leptotene/zygotene is 55%, 41% and 26% at these three ages, respectively (*Goetz et al., 1984*). To obtain a more enriched proportion of leptotene/zygotene spermatocytes, germ cell development was synchronized *in vivo* by manipulating the retinoic acid metabolism, as described in *Romer et al., 2018*. Briefly, at day two post-partum, mice were treated daily (by pipette feeding) with WIN 18,446 (100µg/gram of body weight), an inhibitor of retinoic acid synthesis that blocks the differentiation of spermatogonia and thus meiosis entry (*Hogarth et al., 2013*). After 8 to 10 days of treatment, meiosis was initiated synchronously by a single intraperitoneal injection of 100 µg of retinoic acid in 10 µL of DMSO. Between 8 and 9 days after the injection, mice were sacrificed and testes were harvested. At this time point, about 80–85% of spermatocytes were at leptotene/zygotene stage, as assessed by SYCP3, SYCP1 and γH2AFX staining on spermatocyte spreads performed using a small proportion of testis tissue. The remaining testis tissue was processed for nuclei purification and FACS sorting.

## Purification of spermatocyte nuclei and FACS sorting

Synchronized decapsulated testes were fixed in 1% formaldehyde for 10 min. After quenching the reaction, tissues were homogenized and cells were lysed by homogenization with a tight fit homogenizer in homogenizing buffer (50 mM sucrose, 25mM KCl, 5 mM MgCl$_2$, 50 mM NH$_4$Cl, 120 mM Tris pH7.4). After centrifugation, cells were resuspended in iodixanol-based Optiprep density gradient solution (Sigma-Aldrich D1556). First, a 50% iodixanol working solution was prepared by diluting the Optiprep density gradient solution in working solution (150 mM KCl, 30 mM MgCl$_2$, 120 mM Tris pH 7.4) at a ration 1:5. Then, the 50% iodixanol working solution was diluted to a final concentration of 27% in diluent solution (250 mM sucrose, 25 mM KCl, 5 mM MgCl$_2$, 20 mM Tris pH7.4). Resuspended cells were centrifuged at 10,000 g at 4°C for 30 min. After discarding the supernatants, isolated nuclei were labeled in labeling solution (1x Sytox green (Thermo Fisher Scientific, S70020) in 250 mM sucrose, 25 mM KCl, 5 mM MgCl$_2$, 20 mM tris pH7.4, 1% BSA) at room temperature for 2 hr. Labeled nuclei were filtered through a 70 µm cell strainer and FACS-sorted with a BD FACS Melody sorter (100 µm sort nozzle, 2,000–4,000 events/sec, 34 kHz). First, single nuclei were gated based on their light scatter (forward and reverse side scatter) properties. Second, 4C nuclei were gated based on their DNA content assessed through the fluorescence emitted by the Sytox green fixed on DNA observed with the 488 nm laser. Third, 4C nuclei were separated based on light scatter to gate leptotene-zygotene nuclei. Then, about 10 000 sorted nuclei were deposited on each poly-lysine-coated slide and immunostained with anti-SYCP3, -γH2AFX and -SYCP1 antibodies to verify the prophase I stage. Staining conditions and dilutions are the same as described above. Only samples containing ≥90% of nuclei in leptotene and zygotene stage were used for experiments (see below).

## Immunoprecipitation of genomic DNA containing 5-hydroxymethylcytosine (hMeDIP)

hMeDIP was performed using 5 µg of genomic DNA extracted from a population of 1.5 to 2 *10$^6$ leptotene/zygotene spermatocytes (95% pure) (see above). Genomic DNA was obtained by phenol/chloroform extraction and then sonicated to a size of ~150 bp with a Bioruptor pico apparatus (Diagenode, B01060010). Then, Illumina adaptors were added using the NEB Next Ultra Library Preparation Kit (NEB7370S), without the final PCR step. Finally, hMeDIP was performed with the Active Motif hMeDIP Kit (AM, 55010), according to the manufacturers' manual. Enriched fragments were then amplified by PCR using 12 cycles, as recommended by the NEB Next Ultra Library Preparation Kit. Sequencing was performed on a HiSeqX (2 × 150 bp).

## Chromatin immunoprecipitation of PRDM9 and H3K4me3

ChIP experiments were performed with the ChIP-IT High Sensitivity Kit (Active-motif, 53040). Briefly, testes from two or three synchronized mice (see above) were de-capsulated and fixed in complete tissue fixation solution for 10 min. After quenching the reaction, tissues were homogenized, and cell

suspensions prepared by filtering samples through a 40 µm cell strainer. Cells were washed twice with ice-cold 1x PBS, and chromatin was extracted and immunoprecipitated according to the manufacturers' instructions. 30–40 µg of chromatin was used per IP. The following antibodies (amount) were used: affinity purified anti-PRDM9 (4 µg), anti-H3K4me3 (4 µg).

## Quantitative PCR

Immunoprecipitated DNA was quantified using real-time PCR, as described in *Buard et al., 2009*. The immunoprecipitated fraction at all Dom2-specific hotspots (ChIP/Input ratio) was normalized to the immunoprecipitated fraction at the Cast-specific hotspot Hlx1.6, where no PRDM9 or H3K4me3 enrichment is detected in B6 mice that express PRDM9$^{Dom2}$ (*Diagouraga et al., 2018*). As a control for the sample and IP quality, H3K4me3 level was measured at the *Sycp1* promoter. The primer sequences and PCR conditions for the studied sequence tagged sites (STS) (*Pbx1a, 14a, A3, 17b, Hlx1.6, Psmb9.8* and *Sycp1* promoter) were described previously (*Buard et al., 2009*) and are listed in *Supplementary file 3*.

## DMC1 ChIP-SSDS

DMC1 ChIP-SSDS and library preparation were performed as described in *Grey et al., 2017*. Two testes from 5-week-old *Hells$^{fl/-}$* (named *Hells* CTRL in the main text) and three testes from 9-week-old *Hells* cKO mice were used for each replicate. Sequencing was performed on an HiSeq 2500 Rapidmode apparatus (2 × 150 b).

## Next generation sequencing data computational analysis

### Read alignment

After quality control, 5hMeDIP-seq and DMC1 ChIP-SSDS reads were trimmed to 50 bp and filtered to keep the sequencing read quality Phred score > 28. Reads were then mapped to the UCSC mouse genome assembly build GRCm38/mm10. Mapping was done with Bowtie 2 (version 2.3.2) for the 5hMeDIP-seq experiment, using the single-end mode. DMC1 ChIP-SSDS reads were mapped using the previously published tools (*Khil et al., 2012*) that allow dealing with the specificities of this experiment. Only non-duplicated and uniquely mapped reads were kept after all alignments and used for the subsequent analysis.

### Identifying meiotic hotspots using DMC1 ChIP-SSDS data

To identify meiotic hotspots from biologically replicated samples analyzed by DMC1 ChIP-SSDS, the Irreproducible Discovery Rate (IDR) methodology was used, as previously described for this experiment (*Diagouraga et al., 2018*). This method was developed for ChIP-seq analysis and extensively used in the ENCODE and modENCODE projects (*Landt et al., 2012*). The framework developed by Qunhua Li and Peter Bickel's group (https://sites.google.com/site/anshulkundaje/projects/idr) was followed. Briefly, this method allows testing the reproducibility within and between replicates by using the IDR statistics. Following their pipeline, peak calling was performed using MACS version 2.0.10 with relaxed conditions (`-pvalue=0.1 -bw1000 -nomodel -shift400`) for each of the two replicates, the pooled dataset, and pseudo-replicates that were artificially generated by randomly sampling half of the reads twice for each replicate and the pooled dataset. Then IDR analyses were performed, and reproducibility was checked. Final peak sets were built by selecting the top N peaks from pooled datasets (ranked by increasing p values), with N defined as the highest value between N1 (the number of overlapping peaks with an IDR < 0.01, when comparing pseudo-replicates from pooled datasets) and N2 (the number of overlapping peaks with an IDR < 0.05 when comparing the true replicates), as recommended for the mouse genome. Reproducibility between DMC1 replicates was double-checked by testing their peak strength correlation calculated on the peaks recovered after IDR (Pearson's correlation coefficients were: 0.99 and 0.96 for *Hells* CTRL and *Hells* cKO; *Figure 3—figure supplement 1*).

### Comparisons of DSB hotspot maps

All DSB hotspot maps were compared by identifying overlapping (and non-overlapping) peak centers ± 200 bp. First, it was confirmed that the control (i.e. *Hells* CTRL mice) DSB map reflected the wild-type situation, with 96% of *Hells* CTRL DSB hotspots overlapping with the DSB map in B6

mice (*Grey et al., 2017*), and up to 99% when compared with another DSB map in B6 mice (*Smagulova et al., 2016*). Then, the *Hells* CTRL and *Hells* cKO and the *Hells* cKO and *Prdm9* KO DSB maps were compared (*Figure 3A and C*). DSB hotspots and signals were also visually inspected along the genome (a representative view around position 185 Mb of chromosome one is shown in *Figure 3E*).

## Signal normalization and quantitative analysis (DMC1, 5hmC and 5mC)

If not otherwise stated, all read distributions and signal intensities presented in this work were calculated after pooling reads from both replicates and were expressed as read per millions of mapped reads or fragments. DMC1 ChIP-SSDS signal at DSB hotspots was calculated after peak re-centering, and fragment count was normalized to the local background, as previously described (*Brick et al., 2012*), then normalized to the library size (estimated as the sum of type1-ssDNA, type2-ssDNA and dsDNA). As we previously stated (*Papanikos et al., 2019*), normalization between *Hells* CTRL and *Hells* cKO samples could not be computed because of altered DMC1 dynamics in the *Hells* cKO. The 5hMeDIP-seq signal was calculated at different genomic regions (the region type and size are detailed in the figure legends) by subtracting the library-normalized input signal from the library-normalized 5hMeDIP-seq signal. For *Figure 5* and *Figure 5—figure supplement 1C and D*, DMC1-SSDS B6 or RJ2 sites were divided in four groups, containing an increasing number of CpGs (0, 1–2, 3–5, and $\geq$ 6) within a window of +/- 250 bp around the peak center. Besides the group without CpGs, groups were defined to have similar numbers of DMC1-SSDS sites, as follows: 0 CpGs (812 and 821 for the B6 and RJ2 strains respectively); 1–2 CpGs (3915 and 4126 for the B6 and RJ2 strains, respectively); 3–5 CpGs (5649 and 5851 for the B6 and RJ2 strains, respectively); $\geq$6 CpGs (4384 and 4377 for the B6 and RJ2 strains, respectively). The 5mC signal at whole-genome scale, at DSB sites, in LINE, IAPs and ICRs was calculated from whole-genome bisulfite sequencing data from *Gaysinskaya et al., 2018*. (PRJNA326117). After removing adapter contamination and low-quality reads using trim galore, bisulfite-converted reads were mapped to the UCSC mouse genome assembly build GRCm38/mm10. Mapping in a paired-end mode and methylation call was done using Bismark. Duplicates were not discarded. For the subsequent analysis, only regions with at least one CpG and one informative read were considered. Using the Bedtools suite, the DNA methylation ratio was averaged in 1 kb sliding, non-overlapping windows at the whole genome scale and in the whole interval at DSB sites, in LINE, IAP and ICRs. Median values of 5mC were higher than those reported by *Gaysinskaya et al., 2018* and by *Chen et al., 2020*. For instance, at leptonema, we obtained a genome-wide median DNA methylation level of 91% compared to 77% and 81% respectively in these two studies. These differences, which do not alter the conclusions could be due to the procedures used for reads selection and/or quantification.

### Statistical analysis

The statistical analysis of cytological observations was done with GrapPad Prism 7. Statistical tests for DMC1 ChIP-SSDS were done using R version 3.6.0, and for hMeDIP with python 3.7.4. All tests and p-values are provided in the corresponding legends and/or figures.

## Acknowledgements

We thank the following BioCampus Montpellier facilities for providing excellent technical support: the Réseau des Animaleries de Montpellier (RAM) for animal care, the Réseau d'Histologie Expérimentale de Montpellier (RHEM) for histology, and Montpellier Resources Imagerie (MRI) for microscopy, with help from Amélie Sarrazin for FACS. Mass spectrometry experiments were carried out using facilities of the Functional Proteomics Platform of Montpellier. We thank Rhea Kang and Francesca Cole for great teaching and training of M.B to the spermatogenesis synchronization protocol. We thank Christophe Grunau and the epigenomic platform (http://ihpe.univ-perp.fr/plateforme-epi-genetique) for alignments of sequencing data of bisulfite-treated samples and Jihane Basbous for help in HeLa cells verification. This platform is supported with the support of LabEx CeMEB, an ANR 'Investissements d'avenir' program (ANR-10-LABX-04–01). We thank Dan O' Carroll for sending us mice that harbor the Stra8-iCre transgene. We thank Yuki Shinya for performing several yeast two-hybrid assays, Lucile Marion-Poll for advices for FACS on fixed purified nuclei, and James Hutchins for advices on mass spectrometry analysis. BdM was funded by grants from the Centre National

pour la Recherche Scientifique (CNRS), the European Research Council (ERC) Executive Agency under the European Community's Seventh Framework Programme (FP7/2007-2013 Grant Agreement no. [322788]) and MSDAVENIR. BdM was recipient of the Prize Coups d'Élan for French Research from the Fondation Bettencourt-Schueller. MB was funded by a PhD fellowship from Fondation pour la recherche médicale (FRM).

## Additional information

### Competing interests

Bernard de Massy: Reviewing editor, *eLife*. The other authors declare that no competing interests exist.

### Funding

| Funder | Grant reference number | Author |
|---|---|---|
| European Research Council | 322788 | Bernard de Massy |
| MSDAVenir | Gene-IGH | Bernard de Massy |
| Fondation Bettencourt Schueller | | Bernard de Massy |
| Fondation pour la Recherche Médicale | | Mathilde Biot |

The funders had no role in study design, data collection and interpretation, or the decision to submit the work for publication.

### Author contributions

Yukiko Imai, Validation, Investigation, Visualization, Methodology, Writing - review and editing; Mathilde Biot, Data curation, Software, Formal analysis, Validation, Investigation, Visualization, Methodology, Writing - review and editing; Julie AJ Clément, Data curation, Software, Formal analysis, Validation, Visualization, Methodology, Writing - review and editing; Mariko Teragaki, Investigation; Serge Urbach, Data curation, Software, Formal analysis, Investigation, Methodology; Thomas Robert, Investigation, Methodology, Writing - review and editing; Frédéric Baudat, Conceptualization, Investigation, Visualization, Methodology, Writing - review and editing; Corinne Grey, Conceptualization, Supervision, Investigation, Methodology, Writing - review and editing; Bernard de Massy, Conceptualization, Resources, Data curation, Formal analysis, Supervision, Funding acquisition, Validation, Visualization, Methodology, Writing - original draft, Project administration, Writing - review and editing

### Author ORCIDs

Bernard de Massy [ID] https://orcid.org/0000-0002-0950-2758

### Ethics

Animal experimentation: All experiments were carried out according to the CNRS guidelines and were approved by the ethics committee on live animals (project CE-LR-0812 and 1295).

### Decision letter and Author response

Decision letter https://doi.org/10.7554/eLife.57117.sa1
Author response https://doi.org/10.7554/eLife.57117.sa2

# Additional files

## Supplementary files

• Supplementary file 1. List of all the proteins identified by mass spectrometry after purification of protein complexes by immunoprecipitation of PRDM9. Proteins are displayed in four separate sheets: HeLa S3 cell extracts with size selection. Six samples: two from Hela S3 cells that express N-terminally tagged (Nter1 and Nter2) PRDM9, two that express C-terminally tagged (Cter1 and Cter2) PRDM9, and two that do not express PRDM9 (no PRDM9). HeLa S3 cell extracts without size selection. Three samples from Hela S3 cells that express N-terminally tagged (Nter) PRDM, C-termi-nally tagged (Cter) PRDM9, or that do not express PRDM9 (no PRDM9). Mouse testis rep1. Two samples from the IP with the anti-PRDM9 antibody and with rabbit serum (mock). Mouse testis rep2. Four samples: two from the IP with the anti-PRDM9 antibody (PRDM9-1 and PRDM9-2) and two with rabbit serum (mock-1 and mock-2). Proteins are ranked by peptide counts after the PRDM9 IP. Additional quantifications were performed in the mouse testis samples. These include MS/MS count, iBAQ, iBAQ rank difference between PRDM9 IP and mock, LFQ intensity and LFQ intensity rank difference between PRDM9 IP and mock.

• Supplementary file 2. Sequences of the primers used for genotyping.

• Supplementary file 3. Sequences of the primers used for qPCR.

• Transparent reporting form

## Data availability

PRIDE partner repository with the dataset identifier PXD017337. NGS data have been deposited at GEO under series record GSE145768. Source data have been provided for Figure 2C–D and Figure 3G.

The following datasets were generated:

| Author(s) | Year | Dataset title | Dataset URL | Database and Identifier |
|---|---|---|---|---|
| Imai Y, Biot M | 2020 | PRDM9 activity depends on HELLS and promotes local 5-hydroxymethylcytosine enrichment | https://www.ncbi.nlm.nih.gov/geo/query/acc.cgi?acc=GSE145768 | NCBI Gene Expression Omnibus, GSE145768 |
| Imai Y, Biot M | 2020 | Identification of PRDM9 partners | https://www.ebi.ac.uk/pride/archive/projects/PXD017337 | PRIDE, PXD017337 |

The following previously published datasets were used:

| Author(s) | Year | Dataset title | Dataset URL | Database and Identifier |
|---|---|---|---|---|
| Brick K, Thibault-Sennett S, Smagulova F, Lam KG, Pu Y, Pratto F, Camerini-Otero RD, Petukhova GV | 2012 | Extensive sex differences at the initiation of meiotic recombination | https://www.ncbi.nlm.nih.gov/gds/?term=GSE99921 | NCBI Gene Expression Omnibus, GSE99921 |
| Gaysinskaya V, Miller BF, De Luca C, van der Heijden GW, Hansen KD, Bortvin A | 2018 | Transient reduction of DNA methylation at the onset of meiosis in male mice | https://www.ncbi.nlm.nih.gov/bioproject/?term=PRJNA326117 | NCBI BioProject, PRJNA326117 |

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
