## [Decision Letter]

**Acceptance summary:**

PDRM9 spatially controls the meiotic recombination landscape in mammals. This work demonstrates that PDRM9 is recruited by the chromatin remodeller, HELLS. HELLS is required for double strand break formation, and the enrichment of 5-hydroxymethylcytosine at PDRM9 sites. These findings illustrate the complexity of the chromatin environment that influences recombination hotspots.

**Decision letter after peer review:**

Thank you for submitting your article "PRDM9 activity depends on HELLS and promotes local 5-hydroxymethylcytosine enrichment" for consideration by *eLife*. Your article has been reviewed by three peer reviewers, and the evaluation has been overseen by a Reviewing Editor and Jessica Tyler as the Senior Editor The following individuals involved in review of your submission have agreed to reveal their identity: Francesca Cole (Reviewer #1); Saadi Khochbin (Reviewer #3).

The reviewers have discussed the reviews with one another and the Reviewing Editor has drafted this decision to help you prepare a revised submission.

The laboratory of Bernard de Massy first discovered PRDM9 as a meiotic factor that initiates recombination at particular locations in the genome. A series of important publications from this laboratory then revealed the major characteristics of PRDM9 as a regulator of DSB and repair. Among various findings, this team showed that chromatin environment mostly induced by PRDM9 could play an important role in DSB formation and repair. However, why PRDM9-dependent DSBs can promote pairing and progression through meiotic prophase in mouse and PRDM9-independent breaks cannot is a mystery. Here the authors take a proteomic approach to identify PRDM9 partners. They identify that HELLS, a SNF2-like family protein, interacts directly with PRDM9. Using a knockout mouse model, evidence is provided indicating that HELLS seems to be required for the binding of PRDM9 to specific PRDM9 DNA binding sites as well as chromatin in spermatocyte nuclei. They also show that HELLS and PRDM9 are required for accumulation of 5-hydroxymethylcytosine at meiotic hotspots, but SPO11 is not required, suggesting a potential role for this modification in subsequent steps of meiotic DSB repair. This is an important piece of work, that confirms and extends previous studies, however a few points need to be addressed before publication.

Essential revisions:

1) This manuscript was prepared and submitted under particular circumstances, since some major reported findings and conclusions have been recently published by another group (Spruce et al., 2020). It is of note that the major findings of the two reports, the identification of HELLS as a PRDM9 partner and the functional consequences of the complex formation, are concordant. The authors should take advantage of the situation to deeply discuss their data with respect to the data published by Spruce and colleagues. Currently, this comparison is limited to only several brief statements. The recommendation is that in the Discussion section the authors fully develop a discussion on both works, which would increase the interest of this manuscript to the readers. The authors should also give proper credit to this work at appropriate points in the text.

2) The fact that the interaction between HELLS and PRDM9 takes place in mouse testes is critical for the conclusions in this manuscript. However, the authors cite technical difficulties in detecting interaction between HELLS and PRDM9 by western blots after IP in mouse testis. Additional explanation is required here. What kind of technical problems are being alluded to? Since this was demonstrated in the published work (Spruce et al., 2020), their findings should be emphasized here. In addition, the authors claim that DNA is not required for the interaction may need to be revised (subsection “HELLS interacts with PRDM9”, third paragraph) In the PRDM9 IP, HELLS peptides were measured with Mass-spec from two replicates. HELLS peptide counts from PRDM9 IP were 6 and 7, compared to 5 and 1 in the Mock. Is this difference significant? Peptide number may not be a quantitative measurement.

3) The work reported here presents important specific aspects which had not been investigated and reported previously. More specifically, the interesting finding of a role for HELLS in the enrichment of 5hmC at PRDM9 genomic sites is of high interest.

The authors' investigations suggest that a step between PRDM9 binding and DSB formation leads to 5hmC establishment. However, this rather obscure statement needs a deeper and a better discussion. For instance, what do the authors think about a role for DNA methylation at these PRDM9-binding sites? Indeed, a plausible scenario would be that HELLS would mediate a de novo DNA methylation at these sites, which could then be converted into 5hmC by the TET enzymes. Interestingly HELLs is known to interact with members of both TETs and de novo DNA methyltransferases, supporting the above idea. One prediction of this scenario is that the PRDM9-binding sites would be under-methylated at stages preceding the expression and activities of HELLS-PRDM9. Are there publicly available genome methylation data for the authors to test this prediction? If not, it would be interesting that at least the authors fully develop a discussion regarding the origin of the establishment of 5hmC at these sites.

4) HELLS is a generalist chromatin remodelling factor that is required for a variety of chromatin-templated mechanisms, including transcription factor binding and transcription. Therefore, the question arises on how much of the effects observed in *HellscKO* meiotic cells could be indirectly linked to the action of HELLS on genome organization and transcription. Do the authors have an idea on the impact of HELLS on the expression of critical meiotic genes? At least a discussion on this point could be interesting. With respect to this point, DAPI staining seems to show some alteration of DNA staining in the *HellscKO* spermatocytes compared to their control counterparts (i.e., Figure 2B, Figure 2—figure supplement 3B), suggesting the occurrence of global genome organization alterations in the absence of Hells. Is this consistently observed? If yes, could the authors at least highlight and discuss these alterations?

5) The manuscript should be streamlined and data more clearly presented throughout. The Discussion section reads more like a review. Authors should focus more on their own results and evidence and limit speculation of additional functions that have been shown in somatic cells or model organisms.

---

## [Author Response]

Essential revisions:1) This manuscript was prepared and submitted under particular circumstances, since some major reported findings and conclusions have been recently published by another group (Spruce et al., 2020). It is of note that the major findings of the two reports, the identification of HELLS as a PRDM9 partner and the functional consequences of the complex formation, are concordant. The authors should take advantage of the situation to deeply discuss their data with respect to the data published by Spruce and colleagues. Currently, this comparison is limited to only several brief statements. The recommendation is that in the Discussion section the authors fully develop a discussion on both works, which would increase the interest of this manuscript to the readers. The authors should also give proper credit to this work at appropriate points in the text.

We certainly do not want to minimize the work by Spruce et al., 2020. We already cited their work in our first version of the manuscript (Introduction, Results, and three times in the Discussion). We have now added more citations. In the current version of the manuscript, we quote the data by Spruce et al. in the Introduction, Results, and in the revised Discussion where we now have a paragraph “A chromatin remodeler for PRDM9 binding” that summarizes the data from both studies.

2) The fact that the interaction between HELLS and PRDM9 takes place in mouse testes is critical for the conclusions in this manuscript. However, the authors cite technical difficulties in detecting interaction between HELLS and PRDM9 by western blots after IP in mouse testis. Additional explanation is required here. What kind of technical problems are being alluded to? Since this was demonstrated in the published work (Spruce et al., 2020), their findings should be emphasized here.

We specify in the text that the interaction was detected by Spruce et al. (subsection “HELLS interacts with PRDM9”).

We have done several immunoprecipitation assays using anti-PRDM9 and anti-HELLS antibodies (three different antibodies tested) without convincing results. Our protocol (Dignam protocol) for protein extraction is different from that used by Spruce et al., but has the advantage of being nuclear, and is recognized for its quality of protein complex purification. Note that we used this extraction protocol to prepare HeLa cell extracts that allowed identifying HELLS/PRDM9 interaction in the Co-IP mass spec experiments. Another difference is the anti-HELLS antibodies. However, it is unclear why the anti-HELLS antibodies we tested should be responsible because they work in IP and western blot experiments. Possibly, our conditions were not sensitive enough to detect the signal.

We provide in Author response image 1, an example of one of our negative results of the co-IP, despite efficient immunoprecipitations.

In addition, the authors claim that DNA is not required for the interaction may need to be revised (subsection “HELLS interacts with PRDM9”, third paragraph) In the PRDM9 IP, HELLS peptides were measured with Mass-spec from two replicates. HELLS peptide counts from PRDM9 IP were 6 and 7, compared to 5 and 1 in the Mock. Is this difference significant? Peptide number may not be a quantitative measurement.

We have now included the label-free quantification (LFQ) rank difference derived from the LFQ intensity, which is a more quantitative evaluation of the presence of a protein than peptide counts. LFQ is a normalized score indicating the relative amount of the proteins. From the intensity, a rank is deduced both in the PRDM9 IP and in the mock (IP with IgG). Then the rank difference (rank in IP PRDM9 – rank in IP mock) is calculated. A positive value indicates an enrichment in the PRDM9 IP. No statistical test would be informative on these semi-quantitative analyses. The reproducibility is the best tool we have to validate the conclusion of the interaction between PRDM9 and HELLS. In our study, we performed three independent IPs (mouse testis rep1 and two duplicates in mouse testis rep2). In all three experiments, the LFQ rank difference values of HELLS were positive.

Thus, this is our best validation for the co-immunoprecipitation between PRDM9 and HELLS.

We added the LFQ rank difference in Table 1 because it provides a complementary view of the data in addition to peptide counts.

Concerning the benzonase treatment and its interpretation, we agree with the reviewer and removed the conclusion about the claim of a DNA-independent interaction, but for a different reason. We now think that the interpretation of the results obtained with co-IP experiments treated with benzonase is not straightforward. First, we did not measure how extensive the benzonase treatment was when added to the nuclear extracts. Second, benzonase can digest genomic DNA, but it is not possible to know whether it might also cleave DNA bound by the PRDM9 zinc finger domain; for instance, this DNA may be inaccessible. This is actually often overlooked in studies using benzonase.

Therefore, we deleted the sentence: “The detection of HELLS peptides in the “Mouse testis rep 2”, protein extracts incubated with benzonase prior to IP suggests that DNA or RNA components are not required for the detection of the interaction between PRDM9 and HELLS”.

3) The work reported here presents important specific aspects which had not been investigated and reported previously. More specifically, the interesting finding of a role for HELLS in the enrichment of 5hmC at PRDM9 genomic sites is of high interest.The authors' investigations suggest that a step between PRDM9 binding and DSB formation leads to 5hmC establishment. However, this rather obscure statement needs a deeper and a better discussion. For instance, what do the authors think about a role for DNA methylation at these PRDM9 binding sites? Indeed, a plausible scenario would be that HELLS would mediate a de novo DNA methylation at these sites, which could then be converted into 5hmC by the TET enzymes. Interestingly HELLs is known to interact with members of both TETs and de novo DNA methyltransferases, supporting the above idea. One prediction of this scenario is that the PRDM9-binding sites would be under-methylated at stages preceding the expression and activities of HELLS-PRDM9. Are there publicly available genome methylation data for the authors to test this prediction? If not, it would be interesting that at least the authors fully develop a discussion regarding the origin of the establishment of 5hmC at these sites.

We agree with these comments and our data certainly raises many questions, particularly how a TET enzyme may be recruited, and what is the status of DNA methylation.

We discussed several possibilities, based on literature data about TET recruitment and interactions (in somatic cells); however, in the context of meiosis, we can only speculate (see Discussion).

Concerning DNA methylation, we can analyse DNA methylation and we did it. This new analysis is now included in Figure 5—figure supplement 1B. We took advantage of the 5-methyl cytosine (5mC) kinetics data during spermatogenesis, before, and during meiosis, published by Gaysinskaya et al., 2018. From the GEO datasets, we aligned the reads obtained after sequencing of bisulphite-treated samples using specific alignment tools and computed 5mC enrichment within several regions (protocol described in the Materials and methods). As controls, we monitored genome-wide LINE, IAP and imprinted control regions (male and female imprints).

We then monitored DSB hotspots, the set of hotspots from B6 (PRDM9^Dom2^) and from RJ2 (Prdm9^Cst^). Note that all DNA methylation data are from B6. We observed that before (spermatogonia) and during meiosis (leptotene and pachytene), 5mC methylation level was as high at hotspots (either active: PRDM9^Dom2^, or not: Prdm9^Cst^) as in the genome (around 90%). This high cytosine methylation level at hotspots was also reported in a recent study using a different technique to monitor methylation (Chen et al., 2020). We cite this data (Results and Discussion).

Therefore, in the Discussion, we propose that hotspots with CpGs have high 5mC levels, and thus the substrate for TET to act is potentially available (of course with the caveat that all this is based on population average, and that we do not have information on single cells).

4) HELLS is a generalist chromatin remodelling factor that is required for a variety of chromatin-templated mechanisms, including transcription factor binding and transcription. Therefore, the question arises on how much of the effects observed in HellscKO meiotic cells could be indirectly linked to the action of HELLS on genome organization and transcription. Do the authors have an idea on the impact of HELLS on the expression of critical meiotic genes? At least a discussion on this point could be interesting.

As pointed out by the reviewers, we agree that HELLS might have other functions during meiosis that could affect genome organization and in turn alter transcription. Therefore, it is an important point to consider. The question whether transcription levels are altered in the absence of HELLS has already been partially addressed by previous works:

In somatic cells

Yu and colleagues reported that Hells-deficient murine embryonic fibroblasts show a widespread loss of CG methylation at uniquely mapped genomic regions, including TSSs of protein-coding genes and non-coding RNA genes. However, despite the dramatic changes in methylation at some promoter regions, the relative transcript steady-state levels remain unchanged and silent regions are not de-repressed, as assessed by RNA-seq. Only 2.2% of genes are up- or down-regulated. Genes with a differential expression are involved in cell adhesion and the defence response, the inflammatory response, cell adhesion, chemotaxis, and leukocyte differentiation. Non-coding RNA expression is not affected, but the expression of some repetitive elements, such as satellite sequences, endogenous retroviral elements, and about 60 subclasses of repeat elements including IAP and MMERVK sequences, is increased (Yu at al., 2014).

In meiotic cells

Spruce and colleagues showed that H3K4me3 levels, assessed by ChIP-seq at PRDM9-independent sites, are not significantly different in HELLS-deficient spermatocytes compared with wild-type ells (Spruce et al., 2020).

De La Fuente and colleagues showed that the lack of HELLS in oocytes induces no change in the transcription levels of a subset of meiosis-associated genes (such as Sycp1, Sycp3, Mlh1 and Prdm9), as assessed by RT-qPCR. However, they observed substantial changes in the methylation levels of the minor and major satellites and of the IAP family of transposable elements. They also showed that in the absence of HELLS, IAP elements are de-repressed (De la Fuente et al., 2006).

It should be noted that male meiocytes have unique mechanisms to repress transposable elements (piRNA pathway), which makes it difficult to transfer these conclusions directly from female to male gametocytes.

Our experimental observations (PRDM9-HELLS interaction, lack of PRDM9 binding in absence of HELLS, and displacement of DSBs to default sites) make us favour the hypothesis of a rather direct role of HELLS in the observed phenotype that phenocopies the absence of PRDM9 binding at DSB hotspots. Nevertheless, we agree with the reviewers that on the basis of the observations cited above, it would be very interesting to look at the expression levels of genes and repetitive elements in our *HellscKO* mice. Unfortunately, due to the current Covid19-related situation, we had to stop the breeding of all our mouse strains. We had one frozen sample from the correct genotype, but at least two duplicates are required. Gathering enough material for a well-performed RNA-seq experiment would have taken not less than 6 months, because two mouse generations are needed to get enough mice because in the progenies from the crosses required for these experiments only one in sixteen mice has the correct genotype (male, Hells fl/- with transgene). We are sorry of not being able to provide this RNA-seq analysis for the current manuscript.

However, we discuss the other potential consequences of Hells depletion in the Discussion.

With respect to this point, DAPI staining seems to show some alteration of DNA staining in the HellscKO spermatocytes compared to their control counterparts (i.e., Figure 2B, Figure 2—figure supplement 3B), suggesting the occurrence of global genome organization alterations in the absence of Hells. Is this consistently observed? If yes, could the authors at least highlight and discuss these alterations?

We agree with the reviewers that the selected images could raise such question. The alteration in DAPI-staining in *HellscKO* spermatocytes noted by the reviewers is most likely due to a weaker DAPI staining on one slide (wild-type) during the experimental procedure.

We evaluated this point by examining the DAPI-staining on *Hellsctrl* and *HellscKO* spermatocytes from other experiments (different mice and different slide preparations), and we do not confirm this difference between wt and HellscKO mice. Centric heterochromatin appears often sharper in *HellscKO* nuclei at the zygotene-like stage, which could be explained by the fact that the chromosome axes adopt a pachytene-like structure during the arrest at the zygotene-like stage. Author response image 2 displays several examples of the DAPI staining at different prophase stages in *Hellsctrl* and *HellscKO* spermatocytes.

**Author response image 2. respfig2:**